# Opportunistic Maintenance Strategy for Complex Equipment with a Genetic Algorithm Considering Failure Dependence: A Two-Dimensional Warranty Perspective

**DOI:** 10.3390/s22186801

**Published:** 2022-09-08

**Authors:** Enzhi Dong, Tielu Gao, Zhonghua Cheng, Rongcai Wang, Yongsheng Bai

**Affiliations:** Shijiazhuang Campus of Army Engineering University of PLA, Shijiazhuang, 050003, China

**Keywords:** complex equipment, failure dependence, two-dimensional warranty, opportunistic maintenance, genetic algorithm

## Abstract

Complex two-dimensional warranty equipment is usually composed of many multi-component systems, which include several key components. During the warranty period, conducting maintenance according to the preventive maintenance plan of each component will increase the warranty costs. Opportunistic maintenance is an effective approach to combine the preventive maintenance of each individual component, which can reduce the warranty cost and improve the system availability. This study explored the optimal opportunistic maintenance scheme of multi-component systems. Firstly, the failure rate model and reliability evaluation model of the multi-component system considering failure dependence were established. Secondly, the preventive maintenance plan of each individual component was determined, with the goal of obtaining the lowest warranty cost per unit time in the component life cycle. Thirdly, the preventive maintenance work of each individual component was combined, and the two-dimensional warranty cost model of the multi-component system was established according to the reliability threshold when performing opportunistic maintenance. In the experimental verification and result analysis, the genetic algorithm was used to find the optimal opportunistic maintenance scheme for the power transmission device. The comparative analysis results show that the opportunistic maintenance scheme reduced the warranty cost by 5.5% and improved the availability by 10%, which fully verified the effectiveness of the opportunistic maintenance strategy.

## 1. Introduction

Warranty refers to the maintenance conducted by the manufacturer after equipment has been put into use [1]. With the advancements in science and technology and the continuous development of equipment manufacturing technology, a large amount of new equipment with complex structures is widely used in all aspects of production and life [2]. The systems that make up complex equipment are tightly coupled, with various types of possibilities for failure, frequent changes in equipment status, and high costs. Additionally, the levels and intensities of use are usually high. For example, radar, spacecraft, aircraft, combat vehicles, and ships are all types of complex equipment. This complex equipment plays an important role in national defense, production, and economic activities. Due to the obvious failure dependence between the components of new complex equipment, there are more failure modes, and it is more difficult to determine the failure law [3]. In actual warranty practice, problems of high warranty cost and unguaranteed availability still exist.

New complex equipment is usually composed of many multi-component systems, and warranty activities are usually carried out for individual components constituting multi-component systems. Two-dimensional warranty strategies are widely adopted in new complex equipment, in which one dimension represents calendar time and one dimension represents usage degree (such as driving mileage or usage time). When the warranty period of any dimension reaches the predetermined period, the warranty ends [4]. For example, an electric multiple unit system (such as power-distributed trains driven by electricity) has a two-dimensional warranty period of up to 3 years and 250,000 km. During the two-dimensional warranty period, the manufacturer will also perform preventive maintenance on a regular basis. Preventive maintenance refers to all activities performed to prevent functional equipment failures and keep equipment in a good state through a series of methods, such as failure inspection, condition detection, maintenance, or replacement before equipment failure. Practice has proven that preventive maintenance can effectively reduce the loss caused by unexpected failures of equipment, reduce the cost of maintenance for equipment, and improve the availability of equipment [5]. Zhao et al. [6] presented a new dynamic maintenance strategy, i.e., a condition-based opportunistic maintenance strategy, and applied this strategy to wind turbines, demonstrating the important role of this strategy in saving maintenance costs and improving equipment availability. However, the warranty strategy adopted in this study is one-dimensional, and does not consider the failure dependence between components.

Making reasonable two-dimensional warranty decisions for complex equipment is a common concern of manufacturers and users. Two-dimensional warranty decisions mainly refer to determining a reasonable preventive maintenance scheme to reduce costs within a warranty period. Due to the different failure laws of components constituting a multi-component system, the isolated maintenance of each individual component will lead to excess maintenance and increase the frequency and cost of preventive maintenance. On the other hand, mechanically combining the preventive maintenance work of various components will lead to the advance of the overall preventive maintenance work and waste the use value of equipment [6]. Most of the current research does not consider the failure dependence between multiple components, and there is usually a large error between the maintenance scheme obtained based on this and the optimal scheme [7].

On the basis of existing research, this study expands the one-dimensional warranty to the two-dimensional warranty, and considers the failure dependence between multiple components, thus, broadening the scope of application of the model. The main problems solved in this paper are detailed as follows:This study adopts the opportunistic maintenance strategy and two-dimensional warranty method to determine the replacement cycle of each individual component in the component life cycle with the goal of achieving the lowest warranty cost per unit time.Through opportunistic maintenance, the preventive maintenance work of each individual component is adjusted according to the reliability threshold when carrying out opportunistic maintenance. Then, the opportunistic maintenance plan of the multi-component system is formed.In a case study, a power transmission device is used as an example for analysis. The reliability threshold for opportunistic maintenance of each component is solved using a genetic algorithm. Finally, the effectiveness of the proposed method is verified through comparative analysis.

This article is organized into seven sections. Section 1 introduces the research background, the main focus of this paper, and the problems to be solved. Section 2 systematically summarizes the literature related to this study. Section 3 presents the model description and provides reasonable assumptions. The fourth section details the constructed opportunistic maintenance cost model of the multi-component system during the two-dimensional warranty period, as well as a decision model with objective function and constraints with the goal of minimum cost. The fifth section describes the main steps of the solution algorithm of the model. The sixth section takes the power transmission device as an example, obtains the optimal opportunistic maintenance scheme of the power transmission device, and performs a comparative analysis. Finally, Section 7 presents the conclusions.

## 2. Literature Review

Before making warranty service decisions for a system, the system characteristics should first be analyzed, then the warranty strategy should be determined, and finally, the maintenance strategy should be determined. This paper mainly focuses on the failure dependence of multi-component systems, adopting a two-dimensional warranty strategy, as well as an opportunistic maintenance strategy. Therefore, the literature review was mainly performed in three parts: Section 2.1 introduces the research status of multi-component system failure dependence; Section 2.2 introduces the research status of two-dimensional warranty theory; Section 2.3 introduces the research status of opportunistic maintenance strategies. 

### 2.1. Failure Dependence of Multi-Component

A “system” usually refers to an entity with certain structures and functions composed of a certain number of components or subsystems [8]. System maintenance strategies are based on component maintenance strategies but are very complex because they involve more components and complex structures. The multi-component maintenance model concerns the optimal maintenance strategy of the system. The system includes several groups of units or many components, which may or may not be related [9]. Failure dependence widely exists in multi-component systems; therefore, scholars generally pay more attention to it. Failure dependence mainly refers to the fact that the failure of one component will affect the failure of other related components [10]. The failure dependence between multiple components can be divided into three types [11]. Type I is failure-related [12,13], i.e., when one component fails, it will lead to the failure of other components with a certain probability; type II is failure-rate-related [14,15], i.e., when one component fails, it will increase the failure rate of other components to a certain extent; type III is impact-damage-related [16], i.e., when one component in the system fails, it will cause a certain degree of random damage to other components, and when the random damage accumulates to a certain degree, it will lead to component failure. Many contemporary studies have considered failure dependence in maintenance strategies. Sun [12] introduced the concept of interactive failure, established a quantitative analysis model of failure interactions between components, and presented the derivation method of failure dependence coefficients between components based on experiments, under the branch of early failure dependence research. References [13,14,15,16] all explored the optimal maintenance schemes of failure dependence multi-component systems with the goal of minimizing maintenance costs, but the maintenance strategies adopted are different, including regular inspection [13], imperfect preventive maintenance [14,15], grouping maintenance [15,16], etc. However, few studies have considered failure dependence in warranty decisions. Dong [17] studied the decision-making problem of extended warranties for a wind turbine system based on the cost-effectiveness analysis method. However, this study mainly focuses on the one-dimensional warranty and does not consider the two-dimensional warranty decision-making problem. In their follow-up study, Dong et al. [18] considered the influence of fault correlation in the two-dimensional warranty decisions of multi-component systems. However, the maintenance strategy adopted by Dong et al. was scheduled preventive maintenance, without considering the differences in maintenance time between components. This assumption is too idealistic. Based on the above analysis, the focus of this paper is to make the opportunistic maintenance decisions for multi-component systems on the basis of two-dimensional warranties considering the failure dependence between multiple components.

### 2.2. Two-Dimensional Warranty

Warranty services are contractual relationships established between the manufacturer and the consumer (or user). If a product fails within the warranty period, the manufacturer is responsible for rectifying the product failure. Warranty services can not only guarantee the product performance, but also remove psychological worries of consumers about uncertain product quality. From the perspective of manufacturers, warranty services can not only improve consumers’ satisfaction and recognition of products, but also establish brand image and bring potential new customers; from the perspective of consumers, warranty services can not only protect their own rights and interests, but also reduce the product maintenance costs during the warranty period. Warranty strategies are key indicators to measure the quality of warranty services and are also important factors which influence consumers’ purchase decisions. The feature of two-dimensional warranties is that the warranty period includes two variables: calendar time and usage, which can usually be represented by an area on the two-dimensional plane. The warranty policy can change the shape of the warranty area. Wang [19] studied two-dimensional warranty policies and the corresponding shape of the warranty area in detail. According to the time of warranty activities, two-dimensional warranties can be divided into two-dimensional basic warranties and two-dimensional extended warranties. Two-dimensional basic warranties refer to quality guarantee responsibilities and obligations of free maintenance or replacement by the manufacturer after the new equipment enters service and fails within the two-dimensional warranty range. Two-dimensional basic warranty decisions usually aim for the lowest warranty costs [20,21], and some studies have also taken equipment availability during the warranty period into account [22,23]. More and more studies have focused on the formulation of warranty policies, so that both manufacturers and users can accept the warranty policy by striking an equilibrium for manufacturers and users [22]. Preventive maintenance can prevent failures or serious consequences of failures and reduce the losses caused by failure shutdown; therefore, increasing numbers of studies have focused on preventive maintenance [24]. A two-dimensional extended warranty refers to the subsequent equipment maintenance support conducted by the manufacturer when the basic warranty ends. Users can decide whether to purchase extended warranty services [25]. However, most of the current research ignores the failure dependence between multiple components [26,27,28,29].

### 2.3. Opportunistic Maintenance

Opportunistic maintenance refers to the maintenance activities performed for the maintenance of one component bringing opportunities to perform the preventive maintenance of other components, to carry out some preventive maintenance work in advance. Berg [30] studied the opportunistic maintenance strategy of a two-component system and proved its effectiveness. Van et al. [31] proposed a (*n*, *N*) opportunistic maintenance strategy suitable for a two-component series system. The (*n*, *N*) strategy means that when the running time of a component reaches *N*, preventive replacement is performed. At the same time, it is judged whether the running time of other components reaches *n*; if so, opportunity replacement should be conducted. Zheng et al. [32] proposed an opportunity replacement model for multi-component systems, without considering the maintenance of each component after failure. Duarte et al. [33] studied the optimal preventive maintenance cycle of a multi-component series system by applying the perfect maintenance method. Laggounea [34] and others developed an opportunistic maintenance scheme for a hydrogen press system with continuous multi-component operation using the Monte Carlo simulation method. Su [35] and others studied an opportunistic maintenance strategy of wind turbines, aiming to minimize the total opportunistic maintenance cost; they used the rolling window method to conduct dynamic adjustments of wind turbine maintenance activities so as to obtain the optimal arrangement of preventive maintenance activities for the wind turbine. Xue [36] and others established an opportunistic maintenance decision-making model of multi-component systems on the basis of a single-component preventive maintenance plan, aiming for the lowest maintenance cost and taking the ratio of opportunistic maintenance coefficients to component Birnbaum importance and the availability of the system as constraints. No studies have used opportunistic maintenance strategies in warranty decisions, and there has been no research on considering the failure dependence between multiple components when making opportunistic maintenance decisions. This paper attempts to bridge these two research gaps.

## 3. Model Description and Assumptions

### 3.1. Model Description

In this paper, the reliability index of each component is regarded as the basis of maintenance, and the reliability threshold when carrying out preventive maintenance and the reliability threshold when carrying out opportunistic maintenance are set, corresponding to the condition-based maintenance strategy and opportunistic maintenance strategy, respectively. Firstly, the preventive maintenance plan of each individual component is determined by the condition-based maintenance strategy; then, the preventive maintenance plan of each individual component is adjusted and combined with the opportunistic maintenance strategy. This study took the series multi-component system as the research object. During the two-dimensional warranty period $\left(W_{B},U_{B}\right)$, Component i shall be subjected to preventive maintenance according to interval $T_{ki}$. Preventive maintenance includes imperfect preventive maintenance and preventive replacement. Based on the single-component preventive maintenance plan, the relevant work is adjusted according to the reliability threshold in order to reduce the warranty cost.

### 3.2. Model Assumptions

(1)The initial reliability of the component is 1, i.e., it is put into use from new.(2)Maintenance resources are sufficient, and the situation of waiting for maintenance due to insufficient maintenance resources is not considered.(3)There is only one main failure mode of components; multiple failure modes are not considered.(4)The utilization rate can be obtained from historical data, and the usage habits of the same user remain unchanged during the basic warranty period, i.e., the utilization rate remains unchanged, and the utilization rates of different users obey uniform distribution.(5)When component i runs to the minimum reliability requirements (the reliability threshold when performing preventive maintenance), preventive maintenance shall be carried out for the component: preventive maintenance is imperfect maintenance. It will be replaced when $n_{i}+1$ preventive maintenance is conducted.(6)In cases of failure during the preventive maintenance interval, the corrective maintenance (minimum maintenance) shall be adopted for component i.

## 4. Model Construction

### 4.1. Establishment of Reliability Model

The multi-component system contained *S* components in total. The direct construction approach in the univariate method was used to construct the two-dimensional failure rate model of a single component. Based on the construction method developed by Yun and Kang [37], the failure rate function of component i is as follows:(1)$$\lambda_{i}(t|r)=\theta_{0i}+\theta_{1i}r+\theta_{2i}t^{2}+\theta_{3i}rt^{2}\;1\leq i\leq S$$
where $\theta_{0i}$, $\theta_{1i}$, $\theta_{2i}$, and $\theta_{3i}$ are all unknown parameters in the failure rate function, which need to be obtained through parameter estimation. To determine the parameters of the expression, it is necessary to determine the specific expression of $\lambda_{i}(t|r)$ and the distribution G(r) of the utilization rate by counting the sample failure time t and the utilization degree U(U=rt) during the warranty period. The specific steps are as follows:

**Step 1:** The component sample size is determined as *Z* (the number of samples used for the experiment was *Z*). $v_{i}$ indicates the number of failures of the *i*th (i=1,2,…,Z) component within the warranty period. $t_{i,b}$ and $u_{i,b}$ represent the time and usage of the *i*th component, respectively, when the *b*th ($b=1,2,\ldots ,v_{i}$) function failure occurs. The value of $r_{i}$ is calculated: $r_{i}=u_{i,v_{i}}/t_{i,v_{i}}$,1≤i≤Z.

**Step 2:** All $r_{i}$ values are divided into a total of *M* groups. The *m*th (1≤m≤M) interval is represented by $\left[h_{m-1},h_{m}\right)$, $0=h_{0}<h_{1}<\ldots <h_{M}<\infty$. The histogram of $r_{i}$ is drawn. Then, the probability density function g(r) can be fitted through the histogram of $r_{i}$. The detailed calculation method of g(r) is given in the Appendix A.

**Step 3:**$\bar{h_{m}}$ represents the median value of $\left[h_{m-1},h_{m}\right)$. The set $I_{m}$ represents the failure set of all samples whose utilization rate falls within this interval. The utilization rates of these samples are all approximate to $\bar{h_{m}}$. The failure rate $\lambda (t|\bar{h_{m}})$ with the utilization rate of $\bar{h_{m}}$ can be fitted by the failure data in the set $I_{m}$. Figure 1 shows the case where the fitted failure rate is normally distributed.

**Step 4:** All the failure rate curves fitted under the specific utilization rate are visualized in three-dimensional space; then, a three-dimensional surface can be fitted through these curves, i.e., the two-dimensional failure rate function expression $\lambda_{i}(t|r)$ concerning *t* and *r*.

Preventive maintenance is imperfect maintenance. Imperfect preventive maintenance can reduce the degradation level of components but it will affect the mean degradation rate of components in the future. Do Van and Berenguer [38] demonstrated that imperfect preventive maintenance will accelerate the degradation of components after maintenance. The application of improvement factors can describe the effect of preventive maintenance. Two improvement factor models are currently widely used in the field of maintenance: decreasing age factor and increasing failure rate factor. Malik [39] first applied the decreasing age factor model. In this model, the failure rate function of the component after the kth imperfect preventive maintenance, i.e., in the k+1th preventive maintenance cycle, is considered as:(2)$$\lambda_{(k+1)i}(t|r)=\lambda_{ki}(t+\beta T_{ki}|r)\;0<t<T_{(k+1)i}$$
where 0<β<1 is the decreasing age factor of the components. After preventive maintenance, the actual service life of components is reduced for a period of time, and the repair effect is between good as new and bad.

Nalagawa [40] first applied the component increasing failure rate factor model, and believed that after the *k*th preventive maintenance, the component failure rate function can be expressed as:(3)$$\lambda_{(k+1)i}(t|r)=\alpha \lambda_{ki}(t|r)\;0<t<T_{(k+1)i}$$
where α>1 is the increasing failure rate factor. After preventive maintenance, the initial failure rate of components becomes 0, but the change rate of failure rate function increases.

The initial failure rate of components after preventive maintenance can be calculated using the decreasing age factor, and the increasing failure rate factor can effectively describe the phenomenon that the component failure is becoming increasingly frequent. After combining the two improvement factor models, the impact of imperfect preventive maintenance on the failure rate can be described more accurately, as shown in Figure 2.

It can be seen from Figure 2 that the failure rate function of component *i* after the *k*th preventive maintenance becomes:(4)$$\lambda_{(k+1)i}(t|r)=\alpha \lambda_{ki}(t+\beta T_{ki})\;t\in (0,T_{(k+1)i})$$
where α is the increasing failure rate factor, β is the decreasing age factor, and $T_{ki}$ is the length of the *k*th imperfect preventive maintenance cycle.

Due to the failure dependence between components, the failure chain model is used. In the failure chain model, the node which only affects other components and is not affected by other components is called the failure starting point; a node which is only affected by other components without affecting other components is called the failure end point; nodes that are affected by other components and also affect other components are called failure midpoints. The failure chain model is shown in Figure 3 [41].

In Figure 3, A is the starting point of the failure chain, G and F are the end points of the failure chain, and the other nodes are the midpoint of the failure chain. Each component is numbered, as shown in Table 1.

The failure dependence coefficient is used to describe the failure dependence degree between components; then, the failure dependence coefficient matrix χ is:$$\chi =\left[\begin{array}{lllllll} 0 & 0 & 0 & 0 & 0 & 0 & 0\\ \chi_{21} & 0 & 0 & 0 & 0 & 0 & 0\\ 0 & \chi_{32} & 0 & 0 & 0 & 0 & 0\\ \chi_{41} & 0 & 0 & 0 & 0 & 0 & 0\\ 0 & 0 & \chi_{53} & \chi_{54} & 0 & 0 & 0\\ 0 & 0 & 0 & 0 & \chi_{65} & 0 & 0\\ 0 & 0 & 0 & \chi_{74} & 0 & 0 & 0 \end{array}\right]$$
where $\chi_{ab}$ represents the failure influence coefficient of component B on component A. The failure dependence coefficient can be determined through the following approaches: (1) obtained by probability theory; (2) estimated based on the experience of the designer, manufacturer, and maintenance staff; (3) based on mechanical or dynamic estimation; (4) decision based on laboratory experiments. In a system composed of *S* components, considering the failure dependence between components, the actual failure rate of components is composed of two parts: the inherent failure rate and related failure rate. Then, considering the failure dependence, the actual failure rate of components can be expressed as:(5)$$\lambda_{{}_{i}}^{real}\left(t|r\right)=\{I\}\{\lambda_{i}\left(t|r\right)\}+\{\chi \}\{\lambda_{i}\left(t|r\right)\}$$
where $\lambda_{{}_{i}}^{real}\left(t|r\right)$ is the matrix of S×1, 1≤i≤S, which represents the actual failure rate of each component, I is the *S*-order unit matrix, and $\left\{\lambda_{i}\left(t|r\right)\right\}$ is the matrix of S×1, 1≤i≤S, which represents the inherent failure rate of each component. Then, at the end of the *k*th maintenance cycle, the reliability function of component *i* is:(6)$$R_{{}_{ki}}^{real}(t)=\exp[-\int_{0}^{T_{ki}}\int_{r_{l}}^{r_{u}}\lambda_{{}_{i}}^{real}(t|r)g(r)drdt]$$

### 4.2. Determination of the Preventive Maintenance Interval of a Single Component

When the reliability of components is low, the risk of failure during operation is high; thus, the reliability requirements of components are strict. When the reliability of components is low, preventive maintenance will be conducted, including imperfect preventive maintenance and replacement. The reliability threshold $R_{{}_{\min}}^{i}$ when carrying out preventive maintenance is set. When the reliability of component i is lower than this value, preventive maintenance will be performed. Then, the reliability equation is:(7)$$\left\{\begin{array}{l} \exp[-\int_{0}^{T_{1i}}\int_{r_{l}}^{r_{u}}\lambda_{{}_{1i}}^{real}(t|r)g_{i}(r)drdt]=R_{{}_{\min}}^{i}\\ \exp[-\int_{0}^{T_{2i}}\int_{r_{l}}^{r_{u}}\lambda_{{}_{2i}}^{real}(t|r)g_{i}(r)drdt]=R_{{}_{\min}}^{i}\\ \vdots \\ \exp[-\int_{0}^{T_{ki}}\int_{r_{l}}^{r_{u}}\lambda_{{}_{ki}}^{real}(t|r)g_{i}(r)drdt]=R_{{}_{\min}}^{i} \end{array}\right.$$

By solving the above formula, the preventive maintenance interval $T_{ki}$ of component i can be obtained. At the same time, it can be seen from the above formula that the expected number of failures of component i in each preventive maintenance interval is the same: $-\ln R_{{}_{\min}}^{i}$. It can be seen from the assumption that component i has undergone $n_{i}$ preventive maintenance before replacement, and each replacement means the end of the component life cycle. In the life cycle of component i, the warranty cost per unit time can be expressed as:(8)$$C_{average}^{i}=\frac{n_{i}[S_{fi}(-\ln R_{\min}^{i})+S_{pi}]+S_{fi}(-\ln R_{\min}^{i})+S_{ri}+C_{d}[-\ln R_{\min}^{i}(n_{i}+1)T_{fi}+n_{i}T_{pi}+T_{ri}]}{\sum_{k=1}^{n_{i}}[T_{ki}+T_{pi}+T_{fi}(-\ln R_{\min}^{i})]+T_{(n_{i}+1)i}+T_{ri}}$$

The numerator of Equation (8) represents the warranty cost per unit time during the component life cycle, which is mainly composed of four parts. Here, $n_{i}[S_{fi}(-\ln R_{\min}^{i})+S_{pi}]$ represents the sum of the total corrective maintenance cost and preventive maintenance cost in the first *n_i_* preventive maintenance intervals, $S_{fi}(-\ln R_{\min}^{i})$ represents the corrective maintenance cost in the *n_i_* + 1th preventive maintenance interval, and $S_{ri}$ represents the cost of the component replacement. Replacement means the end of the component life cycle. $C_{d}[-\ln R_{\min}^{i}(n_{i}+1)T_{fi}+n_{i}T_{pi}+T_{ri}]$ represents the total expected downtime loss in the life cycle of the component. The denominator of Equation (8) represents the total expected length of the component life cycle, where $T_{fi}(-\ln R_{\min}^{i})$ represents the downtime of corrective maintenance during the preventive maintenance interval and $T_{(n_{i}+1)i}$ represents the duration of the *n_i_* + 1th preventive maintenance interval. By minimizing the objective function $C_{average}^{i}$, the optimal preventive maintenance times $n_{i}$ of component i in the life cycle can be obtained. In summary, the optimization equation of the single-component preventive maintenance interval is:(9)$$\left\{\begin{array}{l} \min\;C_{average}^{i}\\ s.t.\int_{r_{l}}^{r_{u}}\exp[-\int_{0}^{T_{ki}}\lambda_{{}_{ki}}^{real}(t|r)dt]g_{i}(r)dr=R_{{}_{\min}}^{i}\\ R_{{}_{\min}}^{i}>0,\;n_{i}>0\\ k=1,2,3,\ldots ,n_{i};\;T_{ki}\geq 0 \end{array}\right.$$

By solving Equation (9), the optimal values of $T_{ki}$ and $n_{i}$ are obtained.

### 4.3. Opportunistic Maintenance Strategy of a Multi-Component System

There have been many studies conducted on two-dimensional warranties. During the two-dimensional warranty period, there are many maintenance strategies available. For example, Iskandar and Murthy [42] divided the two-dimensional warranty area into two sub-areas, adopted the minimum maintenance strategy or replacement strategy in the two areas, and compared the advantages and disadvantages of different schemes. Yun and Kang [37] similarly adopted the method of dividing regions, but expanded the two sub-regions into three sub-regions, comprehensively considering imperfect maintenance and minimum maintenance, and determining the optimal maintenance scheme with the goal of minimum cost. For more of the latest two-dimensional warranty research, we refer to references [26,27,28,29]. Without exception, these studies regard the warranty object as a single component or single system, and do not consider the dependence between multiple components or systems. Focusing on the failure dependence between components, this study adopted the opportunistic maintenance strategy to combine the preventive maintenance work of each individual component so as to reduce the warranty cost and improve the system availability.

In this study, the opportunistic maintenance strategy means that when preventive maintenance is carried out for a component at a certain time, multiple components with similar preventive maintenance times are subjected to preventive maintenance as well. Thus, opportunistic maintenance adjusts the maintenance time for certain components and leads to early component maintenance. It is common knowledge that performing maintenance ahead of schedule will reduce the failure risk of components and reduce the maintenance costs caused by component failure, although it will waste a certain use value of components. Therefore, it is necessary to control certain conditions when carrying out opportunistic maintenance. Thus, the reliability threshold when conducting opportunistic maintenance for components is introduced.

The reliability threshold when carrying out opportunistic maintenance for component i is set as $\Delta R_{i}$. It is judged whether the difference between the reliability $R_{i}(t)$ of component i and the preventive maintenance threshold $R_{{}_{\min}}^{i}$ is greater than $\Delta R_{i}$ at time *t*. If it is less than $\Delta R_{i}$, opportunistic maintenance is considered to be performed on component i; otherwise, performing opportunistic maintenance on component *i* is not considered. Comprehensively considering reducing expected failure maintenance costs and waste of use value caused by the premature preventive maintenance, it is determined whether component *i* should be subject to opportunistic maintenance at time *t_3_*, as shown in Figure 4.

In Figure 4a, the difference between the reliability $R_{i}(t_{3})$ of component i and the preventive maintenance threshold $R_{{}_{\min}}^{i}$ is less than $\Delta R_{i}$; thus, opportunistic maintenance is considered at time *t_3_*. In Figure 4b, the difference between the reliability $R_{i}(t_{3})$ of component i and the preventive maintenance threshold $R_{{}_{\min}}^{i}$ is greater than $\Delta R_{i}$; thus, opportunistic maintenance is not considered at time *t_3_*. Assuming a total of *h_i_* opportunistic maintenance procedures conducted for component i during the warranty period, the change in failure maintenance cost caused by the advanced maintenance during the *u_i_*th ($1\leq u_{i}\leq h_{i}$) opportunistic maintenance is:(10)$$\Delta C_{iu_{i}}^{1}=S_{fi}\int_{0}^{\Delta t}\int_{r_{l}}^{r_{u}}\lambda_{{}_{i}}^{real}(t|r)drdt$$

The use value waste caused by the advance of maintenance is:(11)$$C_{iu_{i}}^{2}=C_{ai}[R_{i}(t)-R_{{}_{\min}}^{i}]/(1-R_{{}_{\min}}^{i})$$
where $C_{ai}$ is the reliability utilization value of component *i.* The opportunistic maintenance necessity parameter is $\varepsilon_{i}^{u_{i}}$, and its expression is:(12)$$\varepsilon_{i}^{u_{i}}=\Delta C_{iu_{i}}^{1}-C_{iu_{i}}^{2}$$

In summary, two conditions need to be met for the opportunistic maintenance of component *i* at time *t_3_*:$R_{i}(t_{3})-R_{{}_{\min}}^{i}\leq \Delta R_{i}$ and $\varepsilon_{i}^{u_{i}}>0$. When the number of imperfect preventive maintenance procedures of part i reaches $n_{i}+1$, component *i* should be replaced. After completing opportunistic maintenance, component *i* enters the next preventive maintenance interval until the next time for preventive maintenance of the component is reached. Alternatively, when preventive maintenance is carried out on other components, component *i* is analyzed again as to whether opportunistic maintenance is necessary.

Taking the lowest warranty cost of the multi-component system as the decision-making goal, the optimal reliability threshold $\Delta R=(\Delta R_{{}_{1}}^{*},\Delta R_{{}_{2}}^{*},\ldots ,\Delta R_{{}_{i}}^{*},\ldots ,\Delta R_{{}_{S}}^{*})$ when conducting opportunistic maintenance for each individual component is solved. The higher the reliability threshold when performing opportunistic maintenance, the fewer the preventive maintenance procedures of multi-component systems; however, there will be more corrective maintenance procedures and the downtime of corrective maintenance will increase. Therefore, the corrective maintenance cost will increase and the warranty cost of multi-component systems will increase; with the decrease in reliability threshold when carrying out opportunistic maintenance, the number of preventive maintenance procedures of the multi-component system will gradually increase, and the downtime of preventive maintenance will also increase. Therefore, the cost of preventive maintenance will increase. Additionally, the warranty cost of multi-component systems will increase. Hence, there is an optimal reliability threshold when conducting opportunistic maintenance for each component, and the optimal reliability threshold when carrying out opportunistic maintenance can minimize the multi-component warranty cost.

According to Section 4.2, the life cycle $L_{Ci}$ of component *i* is:(13)$$L_{Ci}=\sum_{k=1}^{n_{i}+1}T_{ki}\;1\leq i\leq S$$

At the same time, the time for the preventive maintenance of component *i* can be obtained:(14)$$t_{i}^{m_{i}}=(t_{i}^{1},t_{i}^{2},t_{i}^{3},t_{i}^{4},\ldots ,t_{i}^{m_{i}})$$
where $m_{i}$ is the total number of preventive maintenance procedures of component *i* during the warranty period, $m_{i}=w_{i}(n_{i}+1)+v_{i},1\leq i\leq S$. Here, $w_{i}$ represents the number of preventive replacements of component i during the warranty period, and $v_{i}$ represents the number of preventive maintenance occurrences of part *i* after the last replacement in the warranty period until the end of the warranty period.

Multi-component systems adopt two-dimensional warranties, with a two-dimensional warranty period of $\left[W_{B},U_{B}\right]$. Different utilization rates $r_{z}$ will lead to changes in the actual warranty period, as shown in Figure 5.

In Figure 5, $r_{1}$ represents the shape parameter of the basic warranty area, i.e., the nominal utilization rate $r_{1}=U_{B}/W_{B}$. When $r_{z}\geq r_{1}$, due to the high utilization rate, the warranty period ends early in the time dimension. At this time, the two-dimensional warranty period of components is $\left(U_{B}/r_{z},U_{B}\right)$. When $r_{z}<r_{1}$, the two-dimensional warranty period of the components is $\left(W_{B},U_{B}\right)$. According to the above analysis, the actual warranty period of parts in the time dimension can be expressed as:(15)$$W_{B}^{real}=\left\{\begin{array}{ll} W_{B} & r_{z}<\frac{U_{B}}{W_{B}}\\ \frac{U_{B}}{r_{z}} & r_{z}\geq \frac{U_{B}}{W_{B}} \end{array}\right.$$

When $r_{z}<U_{B}/W_{B}$, the time dimension of the warranty period is $\left[0,W_{B}\right]$; when $r_{z}\geq U_{B}/W_{B}$, the time dimension of the warranty period is $\left[0,U_{B}/r_{z}\right]$. Then, the number of preventive replacements during the warranty period is:(16)$$w_{i}^{1}=\left\{\begin{array}{l} \left\lfloor \frac{W_{B}}{L_{Ci}}\right\rfloor ,\;1\leq i\leq S,\;r_{z}<U_{B}/W_{B}\\ \left\lfloor \frac{U_{B}}{r_{z}L_{Ci}}\right\rfloor ,\;1\leq i\leq S,\;r_{z}\geq U_{B}/W_{B} \end{array}\right.$$

During the warranty period, the warranty cost of multi-component systems consist of four parts: preventive maintenance cost, corrective maintenance cost, preventive maintenance shutdown loss, and corrective maintenance shutdown loss.

Suppose that *Y* preventive maintenance procedures are carried out in total during the warranty period for a multi-component system, and the time of each preventive maintenance is $t_{sys}=(t^{1},t^{2},t^{3},\ldots ,t^{y},\ldots ,t^{Y})$. Assuming that the *y*th (1 ≤ *y* ≤ *Y*) preventive maintenance of the multi-component system happens to be the *j*th preventive maintenance of component *i*, the downtime of component *i* is:(17)$$T_{pij}^{y}=\left\{\begin{array}{ll} T_{pi} & j\neq n_{i}+1\\ T_{ri} & j=n_{i}+1 \end{array}\right.$$

At the same time, other components are assessed for whether it is necessary to perform preventive maintenance. Taking component *l* as an example, the downtime of component *l* when the *y*th preventive maintenance of the multi-component system is expressed as Equation (18). The preventive maintenance cost of component *l* is expressed as Equation (19).
(18)$$T_{pl}^{y}=\left\{\begin{array}{lll} 0 & R_{l}(t^{y})-R_{\min}^{l}>\Delta R_{l} & \\ T_{pl} & R_{l}(t^{y})-R_{\min}^{l}\leq \Delta R_{l} & \;\mathrm{and}\;\mathrm{the}\;\mathrm{number}\;\mathrm{of}\;\mathrm{preventive}\;\mathrm{maintenance}\;\mathrm{procedures}\;\mathrm{does}\;\mathrm{not}\;\mathrm{reach}\;n_{l}+1\\ T_{rl} & R_{l}(t^{y})-R_{\min}^{l}\leq \Delta R_{l} & \;\mathrm{and}\;\mathrm{the}\;\mathrm{number}\;\mathrm{of}\;\mathrm{preventive}\;\mathrm{maintenance}\;\mathrm{procedures}\;\mathrm{reaches}\;n_{l}+1 \end{array}\right.$$
(19)$$C_{pl}^{y}=\left\{\begin{array}{lll} 0 & R_{l}(t^{y})-R_{\min}^{l}>\Delta R_{l} & \\ C_{pl} & R_{l}(t^{y})-R_{\min}^{l}\leq \Delta R_{l} & \;\mathrm{and}\;\mathrm{the}\;\mathrm{number}\;\mathrm{of}\;\mathrm{preventive}\;\mathrm{maintenance}\;\mathrm{procedures}\;\mathrm{does}\;\mathrm{not}\;\mathrm{reach}\;n_{l}+1\\ C_{rl} & R_{l}(t^{y})-R_{\min}^{l}\leq \Delta R_{l} & \;\mathrm{and}\;\mathrm{the}\;\mathrm{number}\;\mathrm{of}\;\mathrm{preventive}\;\mathrm{maintenance}\;\mathrm{procedures}\;\mathrm{reaches}\;n_{l}+1 \end{array}\right.$$

Then, the downtime of the *y-*th preventive maintenance procedure of the multi-component system is:(20)$$T_{sys}^{y}=\underset{\begin{array}{l} 1\leq i\leq S,1\leq j\leq n_{i}\\ 1\leq l\leq S\;andl\neq i \end{array}}{\max}(T_{pij}^{y},T_{pl}^{y})$$

The total downtime of preventive maintenance of a multi-component system during the warranty period is:(21)$$T_{ptotal}=\sum_{y=1}^{Y}T_{sys}^{y}$$

The total downtime of corrective maintenance of a multi-component system during the warranty period is:(22)$$T_{ftotal}=\sum_{i=1}^{S}T_{fi}[w_{i}^{1}\sum_{k=1}^{n_{i}+1}\int_{0}^{T_{ki}}\lambda_{ki}^{real}(t|r)dt+\sum_{k=1}^{v_{i}^{1}}\int_{0}^{T_{ki}}\lambda_{ki}^{real}(t|r)dt]$$

The warranty cost of a multi-component system during the warranty period is:(23)$$C_{1}=\int_{r_{l}}^{r_{1}}\left\{\sum_{y=1}^{Y}\sum_{l=1}^{S}C_{pl}^{y}+\sum_{i=1}^{S}S_{fi}[w_{i}^{1}\sum_{k=1}^{n_{i}+1}\int_{0}^{T_{ki}}\lambda_{ki}^{real}(t|r)dt+\sum_{k=1}^{v_{i}^{1}}\int_{0}^{T_{ki}}\lambda_{ki}^{real}(t|r)dt]+C_{d}(T_{ptotal}+T_{ftotal})\}dG(r\right)$$

Availability is an important index to measure the proportion of normal working time within the total time of multi-component systems throughout a certain period. Users have high requirements for the availability of multi-component systems. The availability of a multi-component system during the warranty period is:(24)$$A_{1}=\int_{r_{l}}^{r_{1}}\left(1-\frac{T_{ptotal}+T_{ftotal}}{W_{B}^{real}})dG(r\right)$$

Under availability constraints, the opportunistic maintenance model of multi-component systems aiming for the lowest warranty cost during the warranty period $\left[W_{B},U_{B}\right]$ is as follows:(25)$$\begin{array}{l} \min C(\Delta R)=C_{1}\\ s.t.\left\{\begin{array}{ll} A_{1}\geq A_{0} & \\ 0\leq \Delta R_{i}\leq 1-R_{\min}^{i} & \;1\leq i\leq S\\ R_{i}(t_{3})-R_{{}_{\min}}^{i}\leq \Delta R_{i} & \;1\leq i\leq S;0\leq t_{3}\leq W_{B}\\ \varepsilon_{i}^{u_{i}}>0 & \;1\leq i\leq S \end{array}\right. \end{array}$$

Here, $A_{0}$ represents the minimum availability acceptable to users.

The greater the value of reliability threshold $\Delta R_{i}$ when carrying out opportunistic maintenance of component *i*, the greater the possibility of performing preventive maintenance on or the replacement of component *i* in advance. The original preventive maintenance plan of the component is changed, and the preventive maintenance interval will become longer, which increases the probability of unexpected failure of the component. Therefore, the corrective maintenance cost and corrective maintenance downtime loss of the system will become higher. The smaller the value of the reliability threshold $\Delta R_{i}$ when carrying out opportunistic maintenance of component *i*, the more the system will be subject to excess preventive maintenance, which will lead to higher preventive maintenance costs and increased downtime of the system. The reliability threshold $\Delta R_{i}$ when conducting opportunistic maintenance will directly affect the preventive maintenance times and intervals of components. Only a reasonable value of $\Delta R_{i}$ can ensure the lowest maintenance cost of the system. The maintenance cost curve of the system is shown in Figure 6 [43]. By solving the model (25), the optimal reliability threshold when performing opportunistic maintenance of each individual component is obtained:(26)$$\Delta R=(\Delta R_{{}_{1}}^{*},\Delta R_{{}_{2}}^{*},\ldots ,\Delta R_{{}_{i}}^{*},\ldots ,\Delta R_{{}_{S}}^{*})$$

## 5. Solution Algorithm

According to the single-component preventive maintenance interval determined in Section 4.2, the optimal preventive maintenance plan of each individual component is further determined; then, the multi-component system opportunistic maintenance strategy in Section 4.3 is used to obtain the optimal reliability threshold when carrying out opportunistic maintenance of each individual component when the warranty cost of the multi-component system is the lowest. The specific algorithm steps are as follows:

**Step 1:** The utilization rate range of a multi-component system is divided into *b* intervals on average, the probability *P_b_* of each interval is calculated according to the distribution of the utilization rate, and the mean value $r_{average}^{b}$ of this interval is used to replace this interval for calculations. *g(r)* is the probability density function that the utilization rate obeys:(27)$$P_{b}=\int_{x}^{y}g(r)dr;\;r_{average}^{b}=\int_{x}^{y}rg(r)dr$$
where *x* is the lower limit of the utilization rate interval and *y* is the upper limit of the utilization rate interval.

**Step 2:** On the premise that the utilization rate is $r_{average}^{b}$, the actual warranty period is obtained according to Figure 5. Equation (9) is solved to obtain the optimal preventive maintenance times *n_i_* of a single component; then, the preventive maintenance interval $T_{ki}$ of each individual component is obtained according to the preventive maintenance threshold $R_{{}_{\min}}^{i}$.

**Step 3:** The preventive maintenance time $t_{i}^{m_{i}}=(t_{i}^{1},t_{i}^{2},t_{i}^{3},t_{i}^{4},\ldots ,t_{i}^{m_{i}})$ is obtained according to each preventive maintenance interval $T_{ki}$ of a single component. Then, the preventive maintenance time of each individual component is arranged in the order from smallest to largest, and the array $t_{sys}=(t^{1},t^{2},t^{3},t^{a},\ldots ,t^{o})$ can be obtained. *a* is the number of imperfect preventive maintenance procedures performed on the system, and 1≤a≤o. The reliability threshold for when the opportunistic maintenance of each individual component is set and the initialization downtime is 0: *t^0^* = 0, *a* = 1, respectively.

**Step 4:** Preventive maintenance is carried out on component *i* at time *t^a^*, and other components are assessed as to whether it is necessary to conduct opportunistic maintenance (as shown in Figure 4). If component *l* needs opportunistic maintenance, it is further determined whether it needs imperfect opportunity preventive maintenance or opportunity replacement. Equation (20) is used to calculate the downtime $T_{sys}^{a}$ of a multi-component system for this preventive maintenance. Equation (19) is used to calculate the preventive maintenance cost for each component of the system.

**Step 5:** The array $t_{sys}$ is updated. The preventive maintenance time of relevant components conducted in advance at *t^a^* time is deleted. If *t^a +^*
^1^ is deleted, the time after *t^a^* is marked as *t^a +^*
^1^.

**Step 6:** The corrective maintenance downtime of each individual component in the (*t^a−^*^1^, *t^a^*) is calculated, and the corrective maintenance downtime of all components is summed to obtain the corrective maintenance downtime $T_{ftotal}^{a}$ of the multi-component system in (*t^a−^*^1^, *t^a^*).

**Step 7:** The total downtime $T_{down}^{a}$ of the multi-component system in (*t^a−^*^1^, *t^a^*) is calculated. $T_{down}^{a}=T_{ftotal}^{a}+T_{sys}^{a}$. Set *a* = *a* + 1. Return to Step 4 until a>o.

**Step 8:** The corrective maintenance downtime $T_{ftotal}^{o+1}$ of the multi-component system from *t^o^* the end of the warranty period is calculated. Then, the total downtime $T_{total}$ of the multi-component system during the warranty period is calculated. The calculation formula is:(28)$$T_{total}=\sum_{a=1}^{o+1}T_{down}^{a}+T_{ftotal}^{o+1}$$

**Step 9:** The expected warranty cost $C^{b}$ of the multi-component system during the warranty period on the premise that the utilization rate is $r_{average}^{b}$ can be calculated according to Equation (23). The expected availability of the multi-component system during the warranty period $A^{b}$ is:(29)$$A^{b}=(1-\frac{T_{total}}{W_{B}^{real}})P_{b}$$

The expected warranty cost C and the expected availability A of the multi-component system during the warranty period is calculated. The calculation formulae are:(30)$$C=\sum_{b}C^{b}$$
(31)$$A=\sum_{b}A^{b}$$

**Step 10:** The availability is judged as to whether it meets the constraints. If so, the cost value is stored as well as the corresponding reliability threshold when carrying out opportunistic maintenance; if not, the cost value is eliminated as well as the corresponding reliability threshold when performing opportunistic maintenance. The reliability threshold when conducting opportunistic maintenance is changed to find the scheme that minimizes the warranty cost.

The algorithm flow chart is shown in Figure 7. The reliability threshold for when the opportunistic maintenance of each component is needed is different; therefore, this model involves multiple variables and nonlinear global optimization problems. Genetic algorithms are more accurate and effective for solving such problems. Therefore, genetic algorithms are used to solve the reliability threshold $\Delta R=(\Delta R_{1}^{*},\Delta R_{2}^{*},\Delta R_{3}^{*},\Delta R_{4}^{*})$ when carrying out opportunistic maintenance for each component in order to make the warranty cost of a power transmission device the lowest under the availability constraints. This model involves four decision variables, the enumeration method will require a large number of calculations, and the accuracy of the results is not high. Genetic algorithms are a fast and effective solution to this kind of multi-variable optimization problem. Genetic algorithms are optimization algorithms based on the concepts of evolution and natural selection. They simulate the process of natural selection through selection, mutation, crossover, and other operations to screen out the optimal individual. Thus, the calculation time can be greatly reduced and the optimal values of decision variables are obtained faster. According to this model, the steps and pseudo codes of the genetic algorithm are given in Appendix B.

## 6. Experimental Verification and Result Analysis

An example of a power transmission device produced by an engine manufacturer in China illustrates the application procedure of the proposed model. All data are from the manufacturer’s historical data or reliability tests. Except for the parameters included in the failure rate function, other parameters can be obtained directly from the manufacturer. The power transmission device was composed of four key components: the valve train component, the lubrication system component, the fuel supply system component, and the starting system component. The component numbers are shown in Table 2. 

The manufacturer is responsible for the power transmission device warranty. Generally, preventive maintenance is carried out independently for each component according to the failure law of each component. However, the manufacturer responded that this maintenance method may lead to wasted maintenance costs, mainly because the preventive maintenance work times of some components are similar. If this preventive maintenance is conducted at the same time, the total number of preventive maintenance procedures of the system will be greatly reduced, thus, reducing the maintenance cost. In view of this, the opportunistic maintenance strategy proposed in this paper is adopted in this case, and the advantages of opportunistic maintenance are demonstrated through comparative analysis. The logic of this case is shown in Figure 8.

### 6.1. Case Preparation

Each component has a trend in degradation in the two dimensions of calendar time and driving mileage. As the mileage increases, the failure rate of the power transmission device will increase; on the other hand, when the engine does not work for a long time, the phenomenon of dry friction or semi-dry friction is more serious when it is restarted, which will accelerate the wear and degradation of components. Therefore, the calendar time and driving mileage affect the reliability of the power transmission device at the same time. Preventive maintenance should pay attention to the calendar time and driving mileage at the same time. The distribution of the service life of each component is as follows:(32)$$\lambda_{i}(t|r)=\theta_{0i}+\theta_{1i}r+\theta_{2i}t^{2}+\theta_{3i}rt^{2}\;1\leq i\leq 4$$

The parameters of the above formula can be estimated by the method proposed in Section 4.1. The sample size *Z* is 500, and the estimated parameters of the failure rate function of each component are shown in Table 3. The utilization rate of the power transmission device (unit: 10^4^ KM / year) follows the uniform distribution of (0.1 ~ 10). Thus, explicit expressions of *g*(*r*) and *G*(*r*) are:(33)$$g(r)=\frac{10}{99}\;0.1\leq r\leq 10;\;G(r)=\frac{r-0.1}{9.9}\;0.1\leq r\leq 10$$

At the same time, it was known that the power transmission device had a two-dimensional warranty period of 2 years, 2 × 10^4^ KM. During the warranty period, the regular preventive maintenance strategy is adopted for each component of the power transmission device, and the corrective maintenance (minimum maintenance) strategy is adopted for unexpected failures within the preventive maintenance interval. When the preventive maintenance of each component reaches a certain number of times, the preventive replacement is implemented. The increasing age factor and decreasing failure rate factor of each component are the same, both α=1.12 and β=0.12. The minimum availability acceptable to users is 0.6. There is a unidirectional failure dependence between the four components, with the failure chain model shown in Figure 9.

In this case, the failure dependence coefficient is estimated by the manufacturer’s experience. The failure dependence coefficient matrix of the power transmission device is:(34)$$\chi =\left[\begin{array}{llll} 0 & 0 & 0 & 0\\ 0.04 & 0 & 0 & 0\\ 0 & 0.02 & 0 & 0\\ 0 & 0.06 & 0.07 & 0 \end{array}\right]$$

Other parameter settings are shown in Table 3.

### 6.2. Maintenance Plan for Individual Components of Power Transmission Devices

It was known that the power transmission device had a two-dimensional warranty period. A year is calculated as 365 days. Firstly, according to the model detailed in Section 4.2, the trend in warranty cost per unit time in the life cycle of each component with the number of preventive maintenance procedures is calculated, as shown in Figure 10.

As can be seen in Figure 10, the optimal number of preventive maintenance procedures in the life cycle of each component are four times, three times, three times, and two times for components 1, 2, 3, and 4, respectively, and the corresponding minimum warranty costs per unit time are 258.9 CNY/day, 170.4 CNY/day, 218.4 CNY/day, and 86.3 CNY/day, respectively. Take the utilization rate *r* as 1 × 10^4^ KM/year as an example: when opportunistic maintenance is not considered, the preventive maintenance intervals of each individual component for the dimensions of calendar time and mileage are shown in Table 4 and Table 5. According to the preventive maintenance interval of each component, under the condition of the utilization rate of 1 × 10^4^ KM/year, the preventive maintenance time of each component for the dimensions of calendar time and mileage (*U*, *U* = *rt*) can be calculated as shown in Table 6 and Table 7. According to Table 6, the preventive maintenance plan of the power transmission device can be obtained, as shown in Figure 11.

A total of 21 preventive maintenance activities were carried out for the power transmission device. After calculation, the total expected warranty cost of the multi-component system was CNY 535,820 and the availability of the multi-component system was 0.8123. With a high number of preventive maintenance procedures, the preventive maintenance downtime increases, resulting in the waste of warranty cost. It can be seen from Figure 11 that there are many occurrences of preventive maintenance being performed with similar times between components. Therefore, combining them through opportunistic maintenance will greatly reduce the number of preventive maintenance procedures needed, thus, reducing the downtime of preventive maintenance and the warranty cost.

### 6.3. Opportunistic Maintenance Plan for the Power Transmission Device

Firstly, the genetic algorithm proposed in Section 5 is used to solve the reliability threshold when carrying out the opportunistic maintenance of each component. The specific parameter settings of the genetic algorithm are shown in Table 8.

Without considering the opportunistic maintenance, according to the maintenance plan of each individual component detailed in Section 6.2, the total expected warranty cost of the multi-component system is CNY 535,820. The genetic algorithm is used to solve the reliability threshold of each individual component concerning opportunistic maintenance. After 270 iterations, ΔR=(0.23,0.241,0.326,0.152) is obtained with the goal of minimizing the warranty cost. At this time, the total expected warranty cost of the multi-component system is CNY 506,514. The total expected availability of the multi-component system is 0.8917. The total expected warranty costs of maintaining the power transmission device with iterations of the genetic algorithm are shown in Figure 12.

In order to demonstrate the advantages of GA, it was compared with the particle swarm optimization (PSO) algorithm and simulated annealing algorithm (SAA). Next, the PSO algorithm and SAA were used for optimization. For the PSO algorithm, the number of iterations was 270 and the population size was 100. The inertia weight of the PSO algorithm was 0.9, the self-adjustment weight was 1.49, and the social-adjustment weight was 1.49. The initial value of ΔR was set as (0.1, 0.1, 0.1, 0.1). The SAA was set to iterate 270 times per temperature, and the initial temperature was 10,000. The cooling rate was 0.9. 

The minimum warranty cost calculated by the PSO algorithm was CNY 523,685, and the corresponding optimal ΔR was (0.33, 0.341, 0.215, 0.157). The minimum warranty cost calculated by the SAA was CNY 513,468, and the corresponding optimal ΔR was (0.26, 0.271, 0.215, 0.163). Schematic diagrams of algorithm iterations of the PSO algorithm and SAA are shown in Figure 13 and Figure 14, respectively.

Through comparison, it was found that the GA can achieve lower warranty costs, and converges earlier with higher operational efficiency; therefore, it has more advantages in solving this model.

At this time, the maintenance plan of the power transmission device needs to be adjusted. According to the opportunistic maintenance strategy of multi-component systems detailed in Abbreviations, the maintenance plan of power transmission devices is adjusted, as shown in Table 9.

In Table 9, IM represents imperfect preventive opportunistic maintenance, PM represents imperfect preventive maintenance, RM represents preventive replacement, IR represents opportunity replacement, and N represents no maintenance work. According to Table 9, the adjusted preventive maintenance plan of the power transmission device can be obtained, as shown in Figure 15. After adjustment, the number of preventive maintenance procedures of the power transmission device is reduced from 21 to 10.

### 6.4. Comparative Analysis

#### 6.4.1. Comparison between Situations with Opportunistic Maintenance and without Opportunistic Maintenance

Opportunistic maintenance combines preventive maintenance performed at different times, effectively reducing the number of preventive maintenance procedures of a multi-component system, thus, reducing the preventive maintenance downtime of multi-component systems during the warranty period and reducing the shutdown loss. Opportunistic maintenance not only reduces the number of preventive maintenance procedures, but also reduces the minimum number of maintenance procedures. This is mainly because opportunistic maintenance advances the time of preventive maintenance for components, which effectively avoids the occurrence of potential failures. The comparison of power transmission device warranty cost and availability between situations with opportunistic maintenance and without opportunistic maintenance is shown in Figure 16.

As shown in Figure 16, the warranty cost of the power transmission device during the warranty period was CNY 535,820 without opportunistic maintenance. After the opportunistic maintenance measures were adopted, the warranty cost of the power transmission device during the warranty period was reduced to CNY 506,514, exhibiting a decrease of 5.5%. Without considering opportunistic maintenance, the availability of the power transmission device during the warranty period was 0.8123. After the opportunistic maintenance measures were adopted, the availability of the power transmission device during the warranty period was increased to 0.8917, an increase of 10%. Through the comparison, it can be seen that the opportunistic maintenance strategy was effective in reducing the warranty cost and improving availability.

#### 6.4.2. Comparison between Situations Considering Failure Dependence and Not Considering Failure Dependence

When the failure dependence between the components of the power transmission device is not considered, i.e., when the failure dependence coefficient matrix χ is zero, using the same solution process and method as detailed in Section 6.3, the warranty cost of the power transmission device can be obtained as CNY 483,216 and the availability is 0.932. The comparison between situations considering failure dependence and not considering failure dependence is shown in Figure 17.

It can be seen from Figure 17 that after considering the failure dependence between components, the warranty cost increased by 4.8% and availability decreased by 4.3%. Clearly, when ignoring the failure dependence between multiple components, the warranty cost of the power transmission device is lower. This result seems to be more acceptable. However, the failure dependence between components exists objectively, and the assumption of failure independence is unrealistic. The assumption of failure independence will lead to serious analysis errors and decision-making errors, which will reduce the warranty cost expectations from the manufacturer. In actual warranty practice, a warranty plan based on this result will increase the cost risks for the manufacturer.

#### 6.4.3. Comparison between One-Dimensional Warranties and Two-Dimensional Warranties

The warranty period of two-dimensional warranties consists of two dimensions, one of which is calendar time and the other of which is usage. In this example, the warranty period of the power transmission device was 2 years: 2 years × 10^4^ KM. The warranty period of one-dimensional warranties consists of one dimension, which is usually calendar time. When a one-dimensional warranty is adopted, the warranty area changes from a rectangle to an open area with an opening above, which means that the warranty period remains unchanged for 2 years, no matter how the utilization rate changes. Figure 18 shows the warranty areas under different warranty methods.

Figure 18a shows a schematic diagram of the warranty area under a two-dimensional warranty, whereas Figure 18b shows a schematic diagram of the warranty area under one-dimensional warranty mode. Table 10 compares the warranty costs and availability of the two different warranty methods for a power transmission device under different utilization rates. According to Table 10, the changing trend in warranty cost and system availability with utilization rate is visualized, as shown in Figure 19 and Figure 20.

As can be seen from Figure 19 and Figure 20, when a two-dimensional warranty is adopted, with the increase in utilization rate, the warranty cost of the power transmission device increases first and then decreases, and the availability of the power transmission device decreases first and then increases; on the whole, however, there is little change. This is mainly because when the actual utilization rate $r_{z}$ ($r_{l}\leq r_{z}\leq r_{1}$) increases, the corrective maintenance times of the power transmission device will increase with the increase in the utilization rate, which will lead to an increase in the corrective maintenance cost. Finally, the warranty cost increases and the availability decreases; when $r_{1}\leq r_{z}\leq r_{u}$, with the increase in utilization rate, the actual warranty period of the power transmission device in the calendar time dimension will be shortened, so the number of preventive maintenance and corrective maintenance instances will be reduced during the warranty period, resulting in a reduction in the warranty cost and increased availability; when the one-dimensional warranty method is adopted, the warranty cost of the power transmission device increases and the availability decreases with the increase in the utilization rate, mainly because the corrective maintenance times of the power transmission device will increase with the increase in the utilization rate, resulting in a continuous increase in the corrective maintenance cost, finally causing a continuous increase in the warranty cost and decreasing availability, and the difference between the warranty cost and availability corresponding to the one-dimensional warranty method and the two-dimensional warranty method becomes larger and larger, which reflects that, with the increase in the utilization rate, the two-dimensional warranty has more advantages in saving warranty costs and improving system availability.

#### 6.4.4. Comparison between Grouping Maintenance and Opportunistic Maintenance

This paper adopted the opportunistic maintenance strategy to combine the preventive maintenance of each individual component so as to reduce the warranty cost and improve the system availability. In addition to the opportunistic maintenance strategy, the grouping maintenance strategy is another widely used maintenance strategy. By introducing the preventive maintenance benchmark interval *T*_J_, the preventive maintenance time of each component is adjusted to an integral multiple of the preventive maintenance benchmark interval so as to achieve the result of combining the preventive maintenance of each component. Reference [42] adopted the method of grouping maintenance. Compared with opportunistic maintenance, the grouping maintenance strategy does not use the reliability information of components; thus, this method is relatively simple and easy to implement. According to the method in reference [44], the variation in system warranty cost with preventive maintenance benchmark intervals is shown in Figure 21.

It can be seen from Figure 21 that when the preventive maintenance benchmark interval was set to 120 days, the system warranty cost was the lowest, at CNY 610,237, which is 20% higher than that of opportunistic maintenance. The system availability was 0.72, which is 19% lower than that of opportunistic maintenance. The grouping maintenance plan of the multi-component system is shown in Table 11.

It can be seen from Table 11 that the number of preventive maintenance procedures of the system is reduced compared with that of opportunistic maintenance, but the interval between two preventive maintenance times of components is expanded, which leads to the frequent occurrence of unexpected failure and increases the cost and downtime of corrective maintenance. This is also the main reason for the higher warranty cost and lower system availability of grouping maintenance compared with opportunistic maintenance. From this point of view, the opportunistic maintenance strategy can make full use of the reliability information of components, which is more scientific and effective than the grouping maintenance strategy.

## 7. Conclusions

This paper considered an opportunistic maintenance strategy for two-dimensional warranty equipment based on the failure dependence of multiple components with the aim of minimizing the manufacturer’s warranty cost. Taking a power transmission device as an example, the results show that:(1)The two-dimensional warranty cost of the power transmission device is significantly reduced, and the availability is significantly improved after implementing the opportunistic maintenance strategy, which fully verifies the effectiveness of this strategy.(2)The assumption of failure independence will lead to serious analysis errors and decision-making errors, meaning that it is difficult to provide support for the formulation of a warranty scheme.(3)The two-dimensional warranty method helps manufacturers save on warranty costs. With the increase in the utilization rate, the advantages of two-dimensional warranties are more obvious compared with one-dimensional warranties.(4)Compared with grouping maintenance, opportunistic maintenance has more advantages in reducing warranty costs and improving system availability.

In practice, in addition to imperfect preventive maintenance and replacement, the preventive maintenance strategy includes a function inspection strategy and a failure detection strategy. Function detection strategies assume that there is a process of functional degradation of components, and its theoretical basis is the delay time model; the failure detection strategy considers the existence of hidden failure modes of the equipment. Adding these preventive maintenance measures to the two-dimensional warranty service decision-making model will greatly improve the applicability of the model, but the modeling process will be more complex, and it is worthy of further study. In future studies, we will consider adding the concept of performance into two-dimensional warranty services. The core aim is to present the warranty service provider with certain rewards or punishments through evaluations of the warranty effect, so as to improve the enthusiasm of the warranty service provider and improve the warranty effect. Considering that it is highly necessary to purchase extended warranty services for some new complex equipment, it is necessary to study the extended warranties offered. Making decisions on complex two-dimensional warranties for equipment with failure dependence will be the focus of future studies.

## Figures and Tables

**Figure 1 sensors-22-06801-f001:**
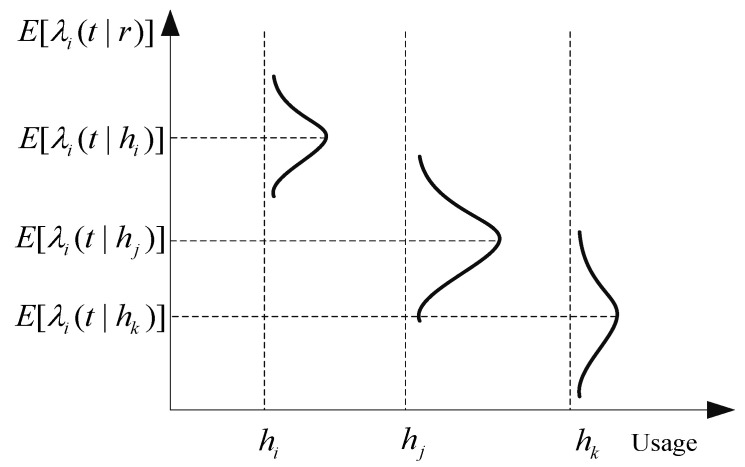
Failure rate function of components under different utilization rates.

**Figure 2 sensors-22-06801-f002:**
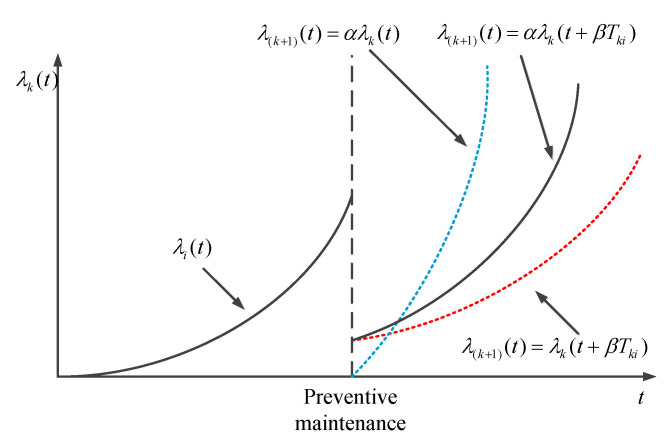
Preventive maintenance effect of components under utilization rate *r*.

**Figure 3 sensors-22-06801-f003:**
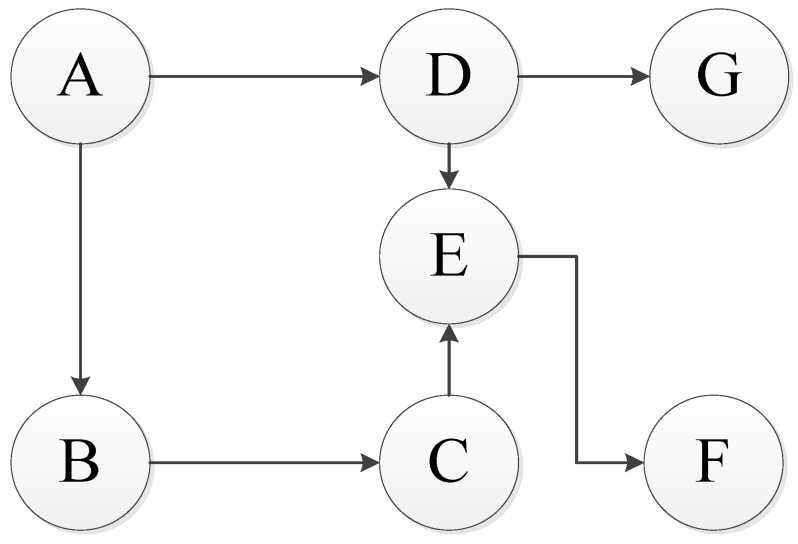
Failure chain model.

**Figure 4 sensors-22-06801-f004:**
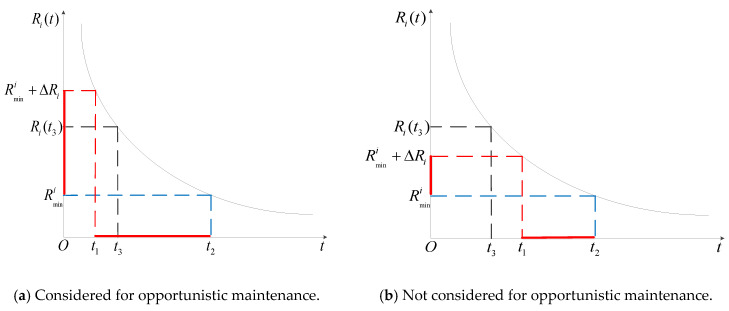
Schematic diagram of component reliability.

**Figure 5 sensors-22-06801-f005:**
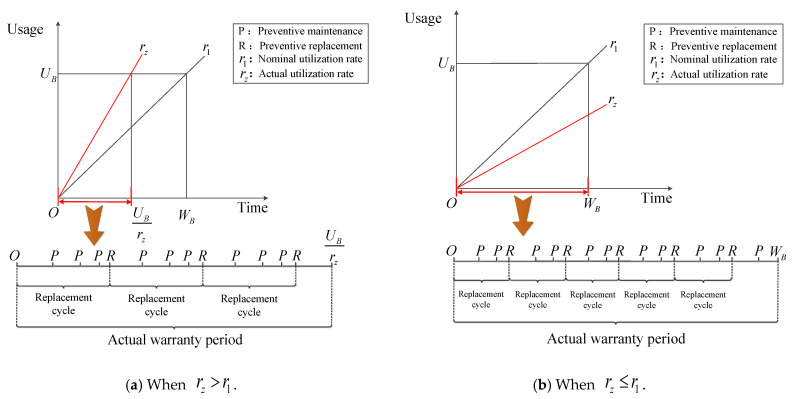
Schematic diagram of the warranty periods under different utilization rates.

**Figure 6 sensors-22-06801-f006:**
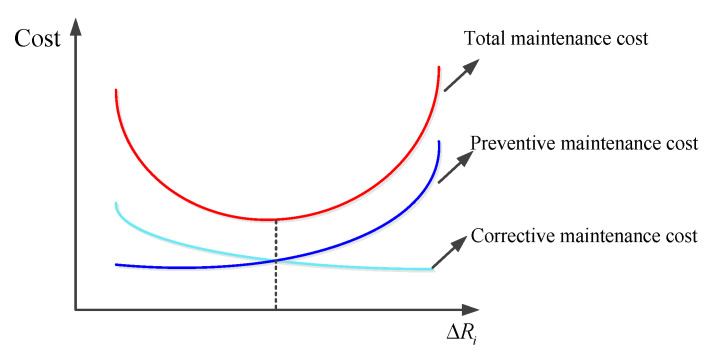
System maintenance cost curve.

**Figure 7 sensors-22-06801-f007:**
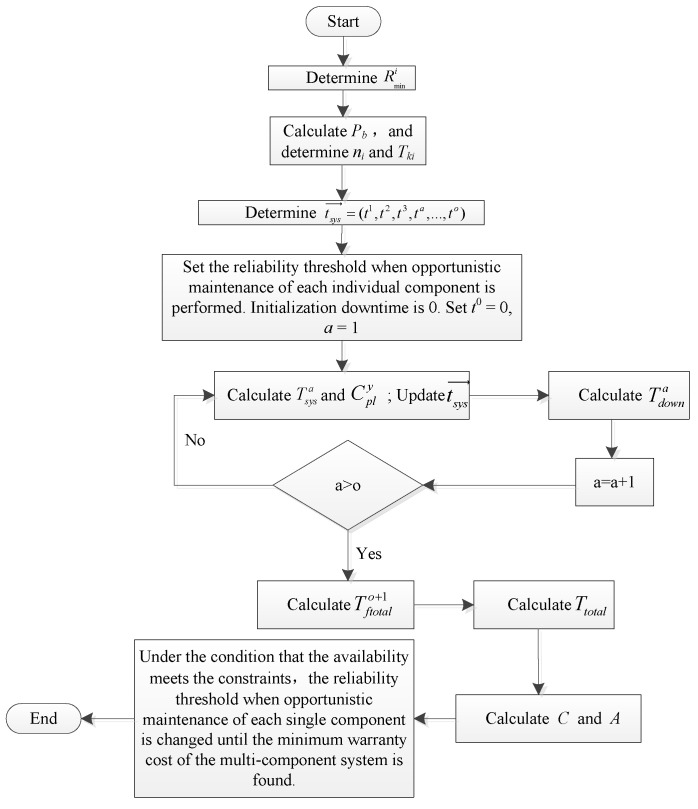
Algorithm flow chart.

**Figure 8 sensors-22-06801-f008:**
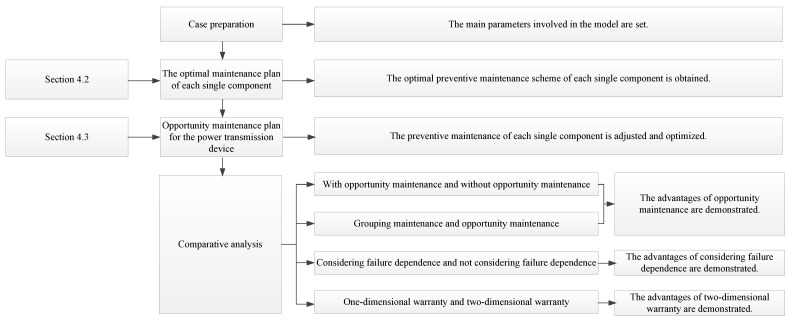
The logic of the case.

**Figure 9 sensors-22-06801-f009:**
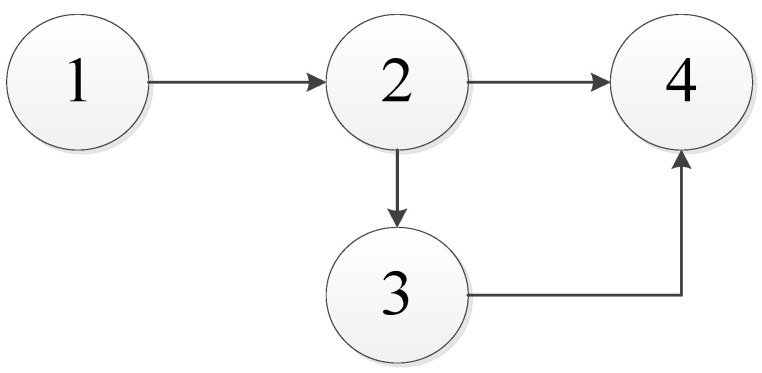
Failure chain model of the power transmission device.

**Figure 10 sensors-22-06801-f010:**
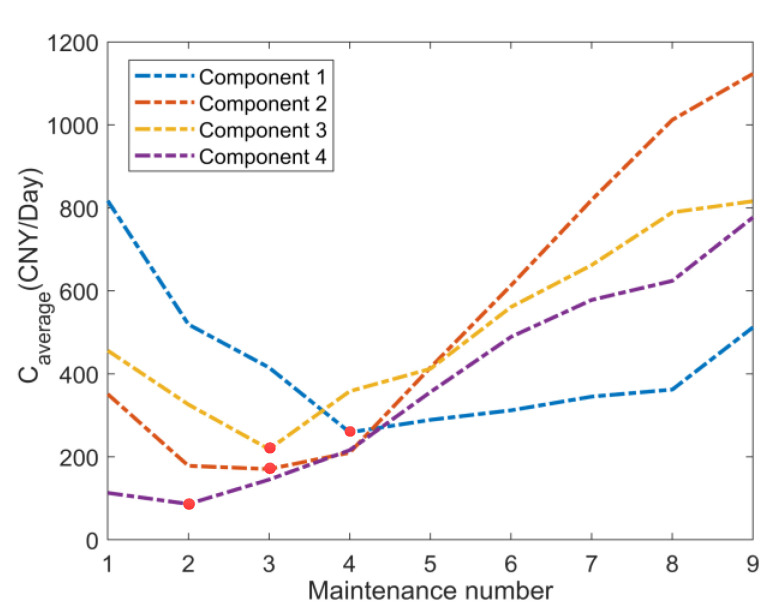
Schematic diagram of warranty cost per unit time changes of each component.

**Figure 11 sensors-22-06801-f011:**
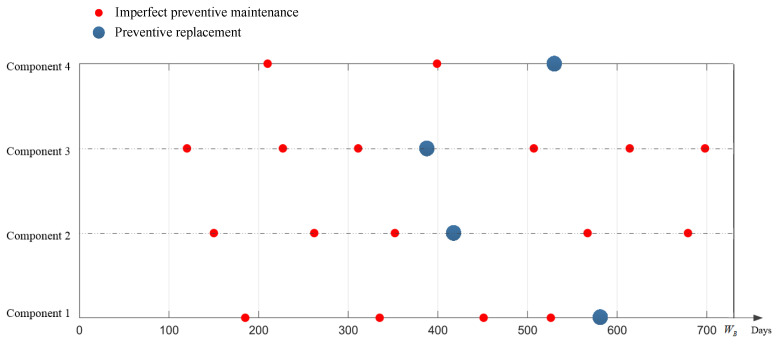
Preventive maintenance plan of the power transmission device.

**Figure 12 sensors-22-06801-f012:**
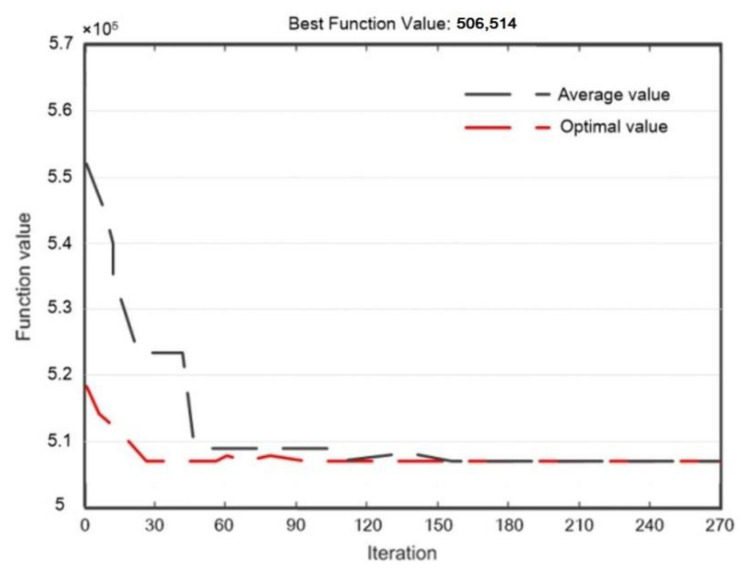
Schematic diagram of the genetic algorithm iterations.

**Figure 13 sensors-22-06801-f013:**
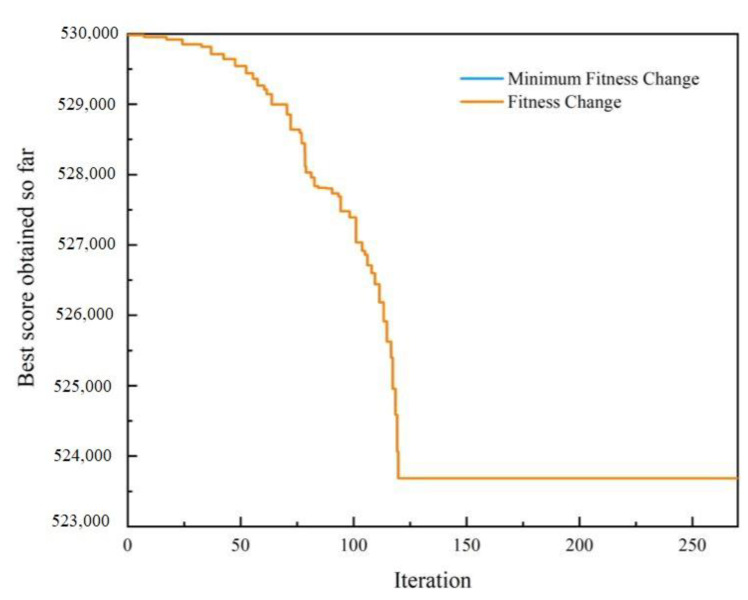
Schematic diagram of iterations of the PSO algorithm.

**Figure 14 sensors-22-06801-f014:**
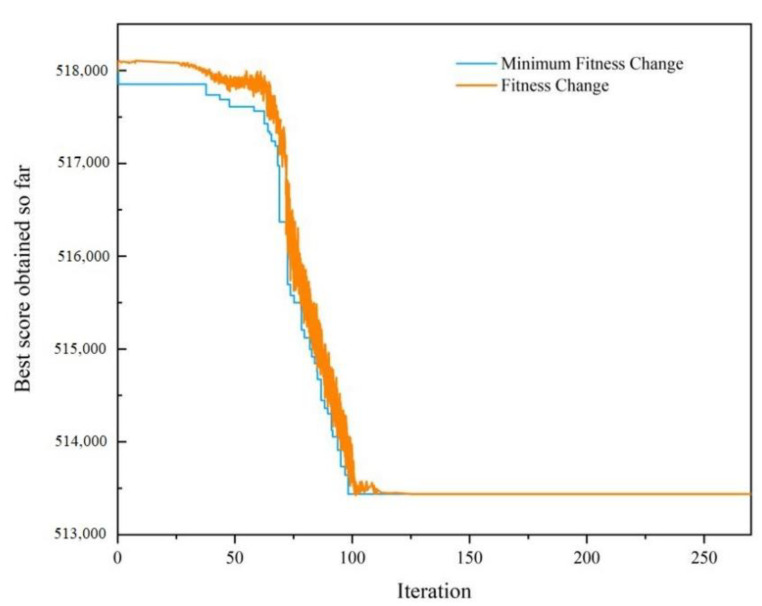
Schematic diagram of iterations of the SAA.

**Figure 15 sensors-22-06801-f015:**
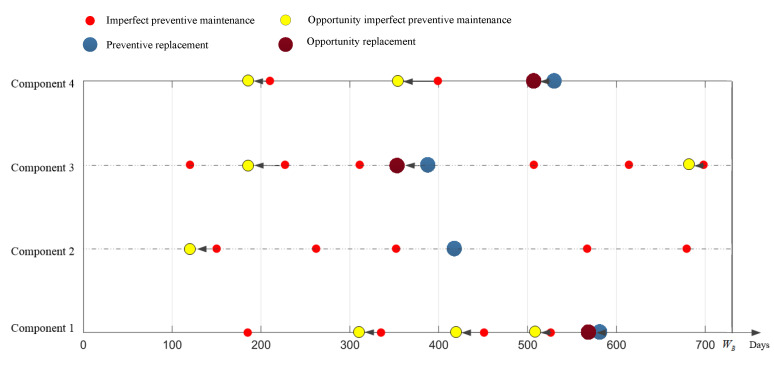
Opportunistic maintenance plan of the power transmission device.

**Figure 16 sensors-22-06801-f016:**
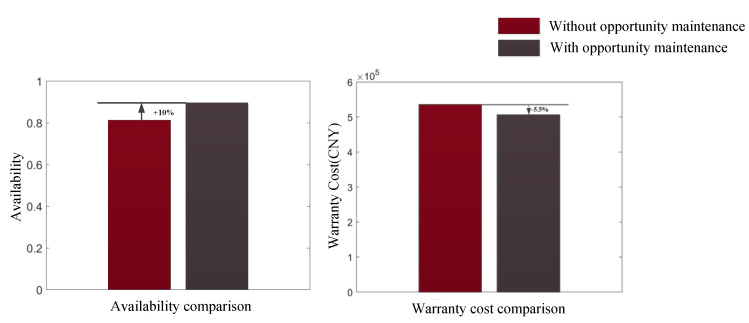
Comparison between situations with opportunistic maintenance and without opportunistic maintenance.

**Figure 17 sensors-22-06801-f017:**
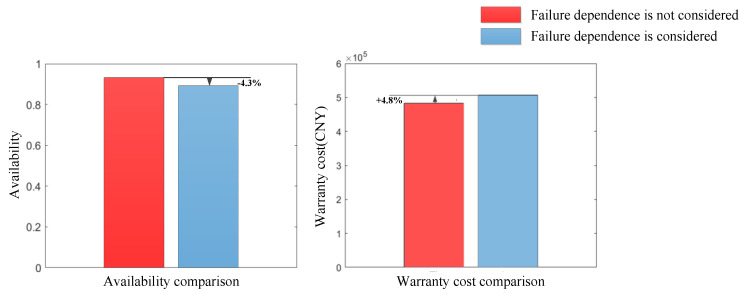
Comparison between situations considering failure dependence and not considering failure dependence.

**Figure 18 sensors-22-06801-f018:**
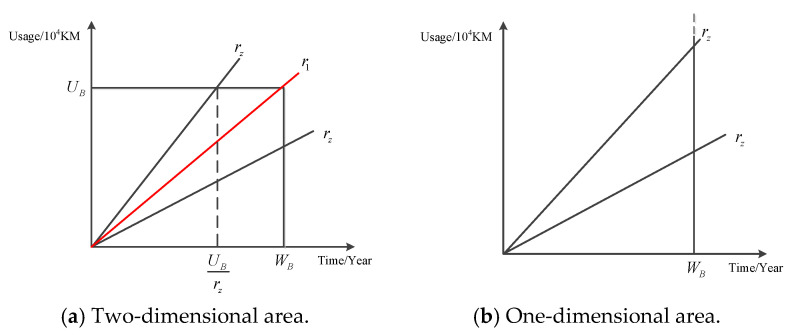
Warranty area under different warranty methods.

**Figure 19 sensors-22-06801-f019:**
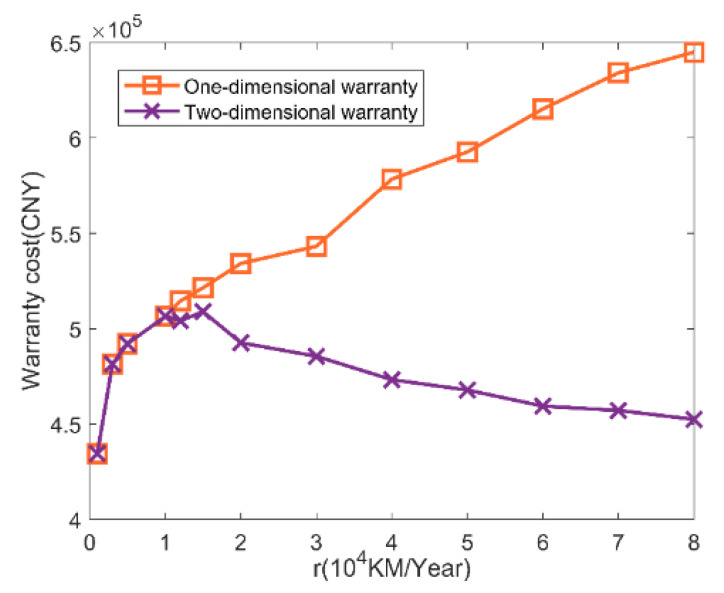
Comparison of warranty cost under different warranty methods.

**Figure 20 sensors-22-06801-f020:**
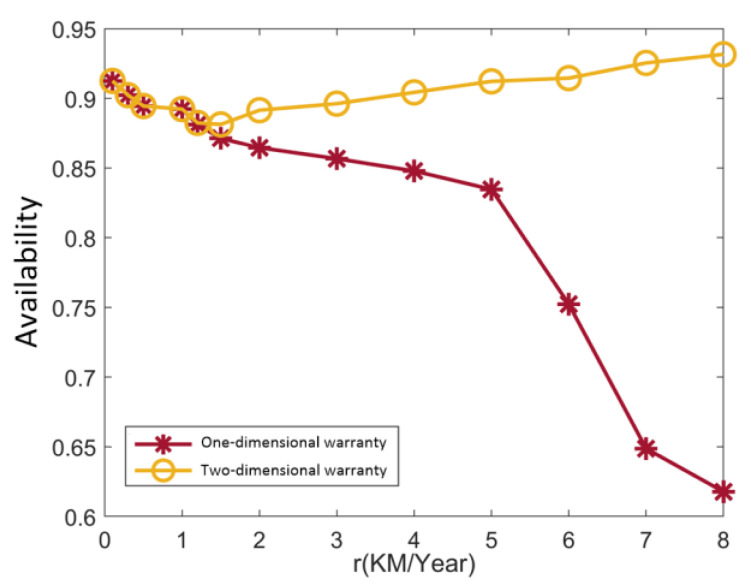
Comparison of availability under different warranty methods.

**Figure 21 sensors-22-06801-f021:**
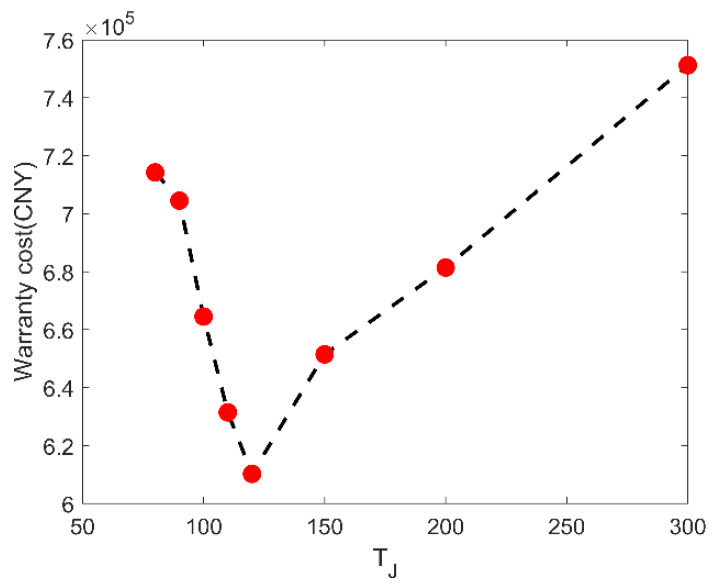
Warranty cost curve under the grouping maintenance strategy.

**Table 1 sensors-22-06801-t001:** Component number.

Component	Number
A	1
B	2
C	3
D	4
E	5
F	6
G	7

**Table 2 sensors-22-06801-t002:** Component numbers of the power transmission device.

Component	Number
The valve train component	1
The lubrication system component	2
The fuel supply system component	3
The starting system component	4

**Table 3 sensors-22-06801-t003:** Parameter settings.

Component	$\theta_{0i}$	$\theta_{1i}$	$\theta_{2i}$	$\theta_{3i}$	CNY			Days			$R_{{}_{\min}}^{i}$	$C_{ai}(\mathrm{CNY})$	$C_{d}(\mathrm{CNY}/\mathrm{Day})$
					$S_{fi}$	$S_{pi}$	$S_{ri}$	$T_{fi}$	$T_{pi}$	$T_{ri}$			
1	0.1	0.06	0.03	0.1	1620	510	140,000	1	2	2	0.3	800	3600
2	0.12	0.05	0.05	0.12	1040	210	113,200	2	3	1	0.2	600	
3	0.08	0.06	0.04	0.08	2510	150	148,600	0.5	1	1	0.2	700	
4	0.07	0.04	0.03	0.07	1450	300	52,410	2	2.5	1.5	0.3	500	

**Table 4 sensors-22-06801-t004:** Preventive maintenance interval of each component (calendar time).

Component	$T_{ki}(\mathrm{Day})$					$C_{average}^{i}\;(\mathrm{CNY}/\mathrm{Day})$	Number of Preventive Maintenance Procedures
	1	2	3	4	5		
1	185	150	116	75	54	258.9	4
2	150	112	90	65		170.4	3
3	120	107	84	76		218.4	3
4	210	189	130			86.3	2

**Table 5 sensors-22-06801-t005:** Preventive maintenance interval of each component (mileage).

Component	Mileage (KM)					$C_{average}^{i}\;(\mathrm{CNY}/\mathrm{Day})$	Number of Preventive Maintenance Procedures
	1	2	3	4	5		
1	5068	4109	3178	2055	1479	258.9	4
2	4110	3068	2466	1781		170.4	3
3	3288	2932	2301	2082		218.4	3
4	5753	5178	3562			86.3	2

**Table 6 sensors-22-06801-t006:** Preventive maintenance times of each component (calendar time).

	Preventive Maintenance Time (Day)							
Component 1	185	335	451	526	580	730		
Component 2	150	262	352	417	567	679	730	
Component 3	120	227	311	387	507	614	698	730
Component 4	210	399	529	730				

**Table 7 sensors-22-06801-t007:** Preventive maintenance time of each component (mileage).

	Preventive Maintenance Time (Mileage)							
Component 1	5068	9178	12,356	14,411	15,890	20,000		
Component 2	4110	7178	9644	11,425	15,534	18,603	20,000	
Component 3	3288	6219	8521	10,603	13,890	16,822	19,123	20,000
Component 4	5753	10,932	14,493	20,000				

**Table 8 sensors-22-06801-t008:** Parameter settings of the genetic algorithm.

Parameters	Value
Population size *d*	50
Elite count	3
Crossover fraction	0.8
Mutation probability	0.01
Stopping criteria	270

**Table 9 sensors-22-06801-t009:** Opportunistic maintenance scheme of the power transmission device.

Time (Day)	Component Maintenance Mode			
	1	2	3	4
120	N	IM	PM	N
185	PM	N	IM	IM
262	N	PM	N	N
311	IM	N	PM	N
352	N	PM	IR	IM
417	IM	RM	N	N
507	IM	N	PM	IR
567	IR	PM	N	N
614	N	N	PM	N
679	N	PM	IM	N

**Table 10 sensors-22-06801-t010:** Comparison of warranty cost and availability under different warranty methods.

Utilization Rates (10^4^ KM/Year)	Warranty Cost (CNY)		Availability	
	One-Dimensional Warranty	Two-Dimensional Warranty	One-Dimensional Warranty	One-Dimensional Warranty
0.1	434,581	434,581	0.9125	0.9125
0.3	481,549	481,549	0.9023	0.9023
0.5	492,146	492,146	0.8945	0.8945
1	506,514	506,514	0.8923	0.8923
1.2	514,526	504,156	0.8812	0.8824
1.5	521,456	508,941	0.8713	0.8813
2	534,219	492,641	0.8645	0.8915
3	543,215	485,549	0.8566	0.8962
4	578,456	473,264	0.8478	0.9043
5	592,694	467,843	0.8346	0.9122
6	615,238	459,418	0.7523	0.9145
7	634,157	457,142	0.6487	0.9254
8	644,963	452,451	0.6178	0.9316

**Table 11 sensors-22-06801-t011:** System preventive maintenance plan under a combined maintenance strategy.

Component	Preventive Maintenance Time (Days)					
	120	240	360	480	600	720
1		•	•	•	•	
2	•	•	•		•	•
3	•	•	•	•	•	•
4		•	•	•		

## Data Availability

All data generated or explored during this study are available in this article.

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
