# Peer review of "Opportunistic Maintenance Strategy for Complex Equipment with a Genetic Algorithm Considering Failure Dependence: A Two-Dimensional Warranty Perspective"

_sensors, 2022, doi:10.3390/s22186801_

Round 1

Reviewer 1 Report

Condition-based opportunity maintenance is an effective way to combine the preventive maintenance of each single component, which can reduce the warranty cost and improve the system availability. This paper explores the optimal condition-based opportunity maintenance scheme of the multi-component system. The comparative analysis results show that the opportunity maintenance scheme reduces the warranty cost by 5.5% and improves the availability by 10%, which fully verifies the effectiveness of the opportunity maintenance strategy. However, there are some problems in the quality of the paper. The specific problems are as follows:

(1)In Introduction, the author should focus on the highlights of the proposed method and what problems have been solved. such as (1)(2)(3).

(2)Many parameters are involved in the proposed method. How to set these parameters?

(3)In 3.3. Notation, Table is recommended.

(4) The conclusion is very bad.

Author Response

Reviewer #1

Concern # 1: In Introduction, the author should focus on the highlights of the proposed method and what problems have been solved. such as (1)(2)(3).

Author response: We thoroughly considered the reviewers’ opinions and made substantial changes to the manuscript.

Author action: We readjusted the structure of the introduction and summarized three highlights of the proposed method and illustrated the solved problems.

(1) This study adopts the opportunistic maintenance strategy and two-dimensional warranty method to determine the replacement cycle of each individual component in the component life cycle with the goal of achieving the lowest warranty cost per unit time.

(2) Through opportunistic maintenance, the preventive maintenance work of each individual component is adjusted according to the reliability threshold when carrying out opportunistic maintenance. Then, the opportunistic maintenance plan of the multi-component system is formed.

(3) In a case study, a power transmission device is used as an example for analysis. The reliability threshold for opportunistic maintenance of each component is solved using a genetic algorithm. Finally, the effectiveness of the proposed method is verified through comparative analysis.

The above contents are added in the introduction, and the changes are indicated with the yellow color.

Concern # 2: Many parameters are involved in the proposed method. How to set these parameters?

Author response: We have fully considered and explained the issue in detail.

Author action: The parameters involved in this paper are mainly shown in Table 3. , ,  and  are unknown parameters in the failure rate function, which should be obtained through parameter estimation. As new content, section 4.1 describes the specific parameter estimation method, highlighted in yellow color. Except for the parameters included in the failure rate function, other parameters can be obtained directly from the manufacturer.

The failure dependence coefficient is also an unknown parameter. The failure dependence coefficient can be determined through the following approaches: (1) obtained by probability theory; (2) estimated based on the experience of the designer, manufacturer, and maintenance staff; (3)based on mechanical or dynamic estimation; (4) decision based on laboratory experiments. In this case, the failure dependence coefficient is estimated by the manufacturer's experience.

The above contents are added in section 6, and the changes are highlighted in yellow. color

Concern # 3: In 3.3. Notation, Table is recommended.

Author response: We thoroughly considered the reviewers’ opinions and made the relevant changes to the manuscript.

Author action: Section 3.3 presents the main notation and meaning in a table format. We also added some main notations and described their meanings to better clarify them.

All modifications are highlighted in yellow color.

Concern # 4: The conclusion is very bad.

Author response: We fully agreed with the reviewers’ opinions and made substantial changes to the conclusion section.

Author action: We rewrite the conclusion. At the same time, several valuable research directions are proposed as future research perspectives.

This paper considered an opportunistic maintenance strategy for two-dimensional warranty equipment based on the failure dependence of multiple components with the aim of minimizing the manufacturer’s warranty cost. Taking a power transmission device as an example, the results show that:

(1) The two-dimensional warranty cost of the power transmission device is significantly reduced, and the availability is significantly improved after implementing the opportunistic maintenance strategy, which fully verifies the effectiveness of this strategy.

(2) The assumption of failure independence will lead to serious analysis errors and decision-making errors, meaning that it is difficult to provide support for the formulation of a warranty scheme.

(3) The two-dimensional warranty method helps manufacturers save on warranty costs. With the increase in the utilization rate, the advantages of two-dimensional warranties are more obvious compared with one-dimensional warranties.

(4) Compared with grouping maintenance, opportunistic maintenance has more advantages in reducing warranty costs and improving system availability.

In practice, in addition to imperfect preventive maintenance and replacement, the preventive maintenance strategy includes a function inspection strategy and a failure detection strategy. Function detection strategies assume that there is a process of functional degradation of components, and its theoretical basis is the delay time model; the failure detection strategy considers the existence of hidden failure modes of the equipment. Adding these preventive maintenance measures to the two-dimensional warranty service decision-making model will greatly improve the applicability of the model, but the modeling process will be more complex, and it is worthy of further study. In future studies, we will consider adding the concept of performance into two-dimensional warranty services. The core aim is to present the warranty service provider with certain rewards or punishments through evaluations of the warranty effect, so as to improve the enthusiasm of the warranty service provider and improve the warranty effect. Considering that it is highly necessary to purchase extended warranty services for some new complex equipment, it is necessary to study the extended warranties offered. Making decisions on complex two-dimensional warranties for equipment with failure dependence will be the focus of future studies.

All modifications are highlighted in yellow color.

Reviewer 2 Report

This paper explores the optimal opportunity maintenance scheme of the multi-component system. The failure rate model and reliability evaluation model of the multi-component system considering failure dependence that the other authors may neglect are established, the preventive maintenance plan of each single component is determined according to the reliability threshold when preventive maintenance of each single component. In addition, by introducing the reliability threshold when opportunity maintenance, the preventive maintenance work of each single component is combined, and the two-dimensional warranty cost model of the multi-component system is established. The innovativeness of this paper meets the requirements of Sensors by and large, but some problems existing in this paper need to be revised carefully:

1.       Most of the papers cited in 2.2 and 2.3 in the Introduction section are papers on maintenance policy, and only [35] is related to warranty. It is not logical to use these literatures to demonstrate the current warranty problem.

2.       The parameters that appear for the first time need to be explained explicitly, such as warranty period (WB, UB), the meaning of WB and UB should be explained directly in 3.3. The subscripts of corrective maintenance downtime and cost should be unified, which is convenient for readers. Thus, the subsequent of life cycle can be replaced with another symbol.

3.       There is no explicit expression for the PDF of g(r).

4.       Tptotal and Tftotal are already expressed in Equations 24 and 25, and there is no need to expand them in Equation 27. A general formula for Equation 25 should be derived that covers w1i and w2i , so that Equations 28 and 29 will be more concise.

5.       Before the numerical case analysis, the appearance of Fig5 that is not interpreted by the references is unreasonable, how to know the change law of the cost curve?

6.       The GA algorithm in 6.2 should be placed in 5 <Solution algorithm>.

7.       There are still some issue about details in the paper:

1) It is recommended to integrate formulas 9 and 10, formulas 11 and 12 into one formula each;

2) usage rate and utilization rate should be unified, and one is chosen as the expression; 3) Throughout the full paper, it can be seen that this article does not involve the current hot field of CBM, this word should be deleted from the title and abstract.

Author Response

Reviewer #2

Concern # 1: Most of the papers cited in 2.2 and 2.3 in the Introduction section are papers on maintenance policy, and only [35] is related to warranty. It is not logical to use these literatures to demonstrate the current warranty problem.

Author response: We thoroughly considered the reviewers’ opinions and made subnational changes to the literature review.

Author action: We have changed the order between sections within the literature review. First, we have explained the logic between the parts in the literature review.

Before making warranty service decisions for a system, the system characteristics should first be analyzed; then, the warranty strategy should be determined; and finally, the maintenance strategy should be determined. This paper mainly focuses on the fail-ure dependence of multi-component systems, adopting a two-dimensional warranty strategy, as well as an opportunistic maintenance strategy. Therefore, the literature re-view was mainly performed in three parts: Section 2.1 introduces the research status of multi-component system failure dependence; Section 2.2 introduces the research status of two-dimensional warranty theory; and Section 2.3 introduces the research status of opportunistic maintenance strategies.

However, few studies have considered failure dependence in warranty decisions. Dong [17] studied the decision-making problem of extended warranties for a wind turbine system based on the cost-effectiveness analysis method. However, this study mainly focuses on one-dimensional warranty and does not consider the two-dimensional warranty decision-making problem. In their follow-up study, Dong et al. [18] considered the influence of fault correlation in the two-dimensional warranty decisions of multi-component systems. However, the maintenance strategy adopted by Dong et al. was scheduled preventive maintenance, without considering the differences in maintenance time between components. This assumption is too idealistic. No studies have used opportunistic maintenance strategies in warranty decisions, and there has been no research on considering the failure dependence between multiple components when making opportunistic maintenance decisions. This paper attempts to bridge these two research gaps.

The above contents are added in section 2, and the changes are highlighted in yellow.

Concern # 2: The parameters that appear for the first time need to be explained explicitly, such as warranty period (WB, UB), the meaning of WB and UB should be explained directly in 3.3. The subscripts of corrective maintenance downtime and cost should be unified, which is convenient for readers. Thus, the subsequent of life cycle can be replaced with another symbol.

Author response: We thoroughly considered the reviewers’ opinions and described the parameters' meaning.

Author action: We detailed the notation and added several important notations. We changed the corrective maintenance cost  for  and the life cycle  for .

All modifications are highlighted in yellow.

Concern # 3: There is no explicit expression for the PDF of g(r).

Author response: We thoroughly considered the reviewers’ comments and made substantial changes to the article.

Author action: In section 6.1, we added the expression for the PDF of g(r). The utilization rate of the power transmission device (unit: 104 KM/year) follows the uniform distribution of (0.1 ~ 10). Thus, explicit expressions of g(r) and G(r) are:

;

All additions are indicated in yellow color.

Concern # 4: Tptotal and Tftotal are already expressed in Equations 24 and 25, and there is no need to expand them in Equation 27. A general formula for Equation 25 should be derived that covers w1i and w2i , so that Equations 28 and 29 will be more concise.

Author response: We fully agreed with the reviewers’ opinions and made substantial changes to the manuscript.

Author action: In the modified manuscript, equation 27 is converted to equation 24. By simplifying equation 24, the general formula of  is deduced. Thus, the unnecessary discussion is reduced, and some formulas are eliminated. In the modified manuscript, formula 27 is more concise.

All modifications are highlighted in yellow.

Concern # 5: Before the numerical case analysis, the appearance of Fig5 that is not interpreted by the references is unreasonable, how to know the change law of the cost curve?

Author response: We thoroughly considered the reviewers’ comments and added more explanations about this issue.

Author action: This conclusion is described in reference [43], which has been cited in the revised version of the manuscript. The reason for the change law of the cost curve is:

       The greater the value of reliability threshold  when carrying out opportunistic maintenance of component i, the greater the possibility of performing preventive maintenance on or the replacement of component i in advance. The original preventive maintenance plan of the component is changed, and the preventive maintenance interval will become longer, which increases the probability of unexpected failure of the component. Therefore, the corrective maintenance cost and corrective maintenance downtime loss of the system will become higher. The smaller the value of the re-liability threshold  when carrying out opportunistic maintenance of component i, the more the system will be subject to excess preventive maintenance, which will lead to higher preventive maintenance costs and increased downtime of the system. The reliability threshold  when conducting opportunistic maintenance will directly affect the preventive maintenance times and intervals of components. Only a reasonable value of  can ensure the lowest maintenance cost of the system. 

All modifications are indicated in yellow.

Concern # 6: The GA algorithm in 6.2 should be placed in 5 <Solution algorithm>.

Author response: We completely considered the reviewers’ opinions and applied the mentioned organization.

Author action: We put the introduction of the genetic algorithm in section 5 and the steps and pseudo-codes of the genetic algorithm in the appendix.

All modifications are highlighted in yellow.

Concern # 7: There are still some issue about details in the paper:

1) It is recommended to integrate formulas 9 and 10, formulas 11 and 12 into one formula each;

2) usage rate and utilization rate should be unified, and one is chosen as the expression; 3) Throughout the full paper, it can be seen that this article does not involve the current hot field of CBM, this word should be deleted from the title and abstract.

Author response: We thoroughly considered the reviewers’ comments and fixed the above issues.

Author action: We integrated formula 9 with formula 10 to construct formula 10 in the revised manuscript. We combined formula 11 with formula 12 to form formula 11 in the revised version of the manuscript. In the entire manuscript, the unified expression of r is the utilization rate. We removed the ‘CBM’ from the title and abstract.

All modifications are indicated in yellow.

Reviewer 3 Report

Dear  authors, the issue you are working on is of highest value and very much appreciated. However, the way you try to present is quite confusing

a) by wording and English

b) replica in the  text

c) very long and undigestable sentences

d) please check whether after ; sign the wording goes on with capital letters or not

e) it would be very good if you tell the reader what is meant by "complex system" - to say : make it clear via an example

f) the way how you come to the coefficient matrix, treating a lubricated drive train and how the lubricant is part of the downtime evaluation is not described and remains dubios.

g) could you probably shift the explanation of your algorithms in an appendix section and write the main text more clear with given real examples ? It would help very much.

h) please check carefully the page numbers. Obviously an error occured from page 24 on by resetting to page nr. 3

Beyond all critism it is to say, that your results are brilliant and very valuable. Even more it should be clear what is meant by real examples as a main topic of the paper.

It is  not clear to me why this is published in Sensors ?

Many thanks anyhow

Author Response

Reviewer #3

Concern # 1: the way you try to present is quite confusing

  1. a) by wording and English
  2. b) replica in the text
  3. c) very long and indigestible sentences.

Author response: We thoroughly considered the reviewers’ comments and made substantial changes to the manuscript.

Author action: We revised the entire manuscript according to the PDF file uploaded by the reviewer and highlighted the modified content in red. Besides, we employed the official editing service of MDPI to modify the English expression of the manuscript. The wording and English of the revised manuscript have improved a lot.

Concern # 2: it would be very good if you tell the reader what is meant by "complex system" - to say : make it clear via an example.

Author response: We thoroughly considered the reviewers’ comments and made substantial changes to the article.

Author action: We illustrated the meaning of complex systems by examples.

       The systems that make up complex equipment are tightly coupled, with various types of failures, frequent changes in equipment status, and high costs. Additionally, the levels and intensities of use are usually high. For example, radar, spacecraft, aircraft, combat vehicles, and ships are all types of complex equipment. This complex equipment plays an important role in national defense, production, and economic activities.

The above contents are added in section 1, and the changes are highlighted in yellow.

Concern # 3: The way how you come to the coefficient matrix, treating a lubricated drive train and how the lubricant is part of the downtime evaluation is not described and remains dubious.

Author response: We thoroughly considered the reviewers’ comments and made substantial changes to the manuscript.

Author action: We added the method in section 4.1 to obtain the failure dependence coefficient.

       The failure dependence coefficient can be determined through the following approaches: (1) obtained by probability theory; (2) estimated based on the experience of the designer, manufacturer, and maintenance staff; (3)based on mechanical or dynamic estimation; (4) decision based on laboratory experiments. In this case, the failure dependence coefficient is estimated by the manufacturer's experience.

       We added Figure 8 in section 6. Figure 8 describes the logic of the case in detail and shows the main contents of the case and the relationship between case analysis and the model construction.

All modifications are highlighted in yellow.

Concern # 4: Could you probably shift the explanation of your algorithms in an appendix section and write the main text more clear with given real examples? It would help very much.

Author response: We thoroughly considered the reviewers’ comments and made substantial changes to the manuscript.

Author action: We put the steps and pseudo-code of the genetic algorithm in the appendix. Section 5 mainly describes the solution ideas based on the proposed model and employs the genetic algorithm to solve this model.

All modifications are indicated in yellow.

Concern # 5: Please check carefully the page numbers. Obviously an error occurred from page 24 on by resetting to page nr. 3.

Author response: We thoroughly considered the reviewers’ comments and made substantial changes to the manuscript.

Author action: We updated the page number. Thank you for the tips.

Concern # 6: It should be clear what is meant by real examples as a main topic of the paper. It is not clear to me why this is published in Sensors ?

Author response: We thoroughly considered the reviewers’ comments and made substantial changes to the manuscript.

Author action: We added figure 8 to section 6. Figure 8 describes the logic of the case in detail, indicating the main contents of the case and the relationship between case analysis and the model construction. As a result, the theme of the article is clarified.

       The manuscript is submitted to the special issue "Condition Monitoring of Mechanical Transmission Systems" of Sensors. The special issue believes that: it is necessary to monitor the health status of mechanical transmission systems to schedule proper maintenance strategies in advance, ensuring the safe operation of machinery and significantly improving industry practices. The special issue's theme is mainly the condition monitoring and maintenance of mechanical transmission devices. The manuscript mainly studies the opportunity maintenance strategy of power transmission devices and considers the failure dependence between various components. The research results are crucial for formulating the maintenance plan of power transmission devices. Thus, this paper can be suitable for publication in the special issue. Before submission, we contacted the guest editor, who confirmed the compatibility of the paper with the special issue's theme.

Reviewer 4 Report

This paper attempts to develop a two-dimensional condition-based opportunity maintenance scheme of the multi-component system with failure dependency.  A genetic algorithm is used to solve the optimization problem. A real case study is conducted to validate the model. I have some major concerns as follows:

1. The paper writing is awful and hard to read. Even in the paper title and abstract, there are several grammar errors. For example, in the title please check “for the failure dependence two-dimensional warranty equipment”. In the abstract, please check “according to the reliability threshold when preventive maintenance of each single component.” and “by introducing the reliability threshold when opportunity maintenance”. Many more grammar issues across the paper exist.

2. The paper organization needs improvement, especially in lack of logic coherence (for example, in Section 2). Can Sections 3 and 4 be combined?

3. There are many model assumptions, e.g., through Equations (1) to (4). What are the real data to validate them? And how? The case study does not explain and just give the parameters. I believe the model parameter estimation itself is a challenging task.

4. The two dimensions (two thresholds) are optimized using a 2-step way, why not using a 1-step procedure to optimize both simultaneously? In the literature (by simply google “opportunistic maintenance, or opportunistic CBM”), many papers optimize them in 1-step.

5. Why must the genetic algorithm be used to solve the problem? This should be explained in detail and would help judge the paper’s contribution.

Author Response

Reviewer #4

Concern # 1: The paper writing is awful and hard to read. Even in the paper title and abstract, there are several grammar errors. For example, in the title please check “for the failure dependence two-dimensional warranty equipment”. In the abstract, please check “according to the reliability threshold when preventive maintenance of each single component.” and “by introducing the reliability threshold when opportunity maintenance”. Many more grammar issues across the paper exist.

Author response: We thoroughly considered the reviewers’ comments and made substantial changes to the manuscript.

Author action: We changed the title to: “Opportunistic maintenance strategy for complex equipment with a genetic algorithm considering failure dependence: a two-dimensional warranty perspective”. We have fixed the grammatical errors in the abstract pointed out by the reviewer. Besides, we contacted the official editing service of MDPI to improve the English expression of the manuscript. The wording and English of the revised manuscript have improved a lot.

Concern # 2: The paper organization needs improvement, especially in lack of logic coherence (for example, in Section 2). Can Sections 3 and 4 be combined?

Author response: We thoroughly considered the reviewers’ comments and made substantial changes to the manuscript.

Author action: There is an inherent logical relationship between the parts of section 2. Before making warranty service decisions for a system, the system characteristics should first be analyzed; then, the warranty strategy should be determined; and finally, the maintenance strategy should be determined. This paper mainly focuses on the failure dependence of multi-component systems, adopting a two-dimensional warranty strategy, as well as an opportunistic maintenance strategy. Therefore, the literature review was mainly performed in three parts: Section 2.1 introduces the research status of multi-component system failure dependence; Section 2.2 introduces the research status of two-dimensional warranty theory; and Section 2.3 introduces the research status of opportunistic maintenance strategies.

       After careful argumentation, we think section 3 is better separated than section 4. Firstly, most papers about maintenance modeling are written in this way. See Junkai Sun et al.(2021, https://doi.org/10.1007/s00170-021-07752-6), Fangqi Dong et al.(2021, http://doi.org/10.1155/2021/5576455), Ali Salmasnia and Maryam Baratian(2021, http://doi.org/10.1080/16843703.2021.1910188), Aiping Jiang et al.(2021, http://doi.org/10.1108/JQME-12-2020-0128). Secondly, if sections 3 and 4 are merged, there will be too many sections in this part, which is not appropriate from the perspective of the whole article. Finally, since section 3 is the basis for the model establishment, it should be presented as a single section to highlight its importance.

Concern # 3: There are many model assumptions, e.g., through Equations (1) to (4). What are the real data to validate them? And how? The case study does not explain and just give the parameters. I believe the model parameter estimation itself is a challenging task.

Author response: We thoroughly considered the reviewers’ comments and made substantial changes to the manuscript.

 In equation (1), , ,  and are unknown parameters in the failure rate function, which should be obtained through parameter estimation. We added the parameter estimation method in the failure rate function in section 4.1. Except for the parameters included in the failure rate function, other parameters can be obtained directly from the manufacturer.

The failure dependence coefficient can be determined through the following approaches: (1) obtained by probability theory; (2) estimated based on the experience of the designer, manufacturer, and maintenance staff; (3)based on mechanical or dynamic estimation; (4) decision based on laboratory experiments. In this case, the failure dependence coefficient is estimated by the manufacturer's experience.

All modifications are indicated in yellow.

Concern # 4: The two dimensions (two thresholds) are optimized using a 2-step way, why not using a 1-step procedure to optimize both simultaneously? In the literature (by simply google “opportunistic maintenance, or opportunistic CBM”), many papers optimize them in 1-step.

Author response: We thoroughly considered the reviewers’ comments and made substantial changes to the manuscript. We agree with the reviewers. Optimizing opportunity maintenance decisions is mostly a one-step approach, such as Aiping Jiang et al.(2021, http://doi.org/10.1108/JQME-12-2020-0128) and Liu Qinming et al.(DOI: 10.3390/EN15020625). The relevant literature indicated that determining inspection points is a critical step in the decision-making optimization of opportunistic maintenance. This paper determines a single component's maintenance plan and the system's opportunity maintenance scheme.  Although it looks like a two-step optimization, it is a one-step optimization. This is mainly because the determination of the single component maintenance plan determines the inspection points, and whether opportunity maintenance should be performed on other components at the inspection point to achieve the model optimization goal. Formula 9 is the core of this paper.

Concern # 5: Why must the genetic algorithm be used to solve the problem? This should be explained in detail and would help judge the paper’s contribution.

Author response: We thoroughly considered the reviewers’ comments and made substantial changes to the manuscript.

Author action: Genetic algorithms are a fast and effective solution to this kind of multi-variable optimization problem. Genetic algorithms are optimization algorithms based on the concepts of evolution and natural selection. They simulate the process of natural selection through selection, mutation, crossover, and other operations to screen out the optimal individual. Thus, the calculation time can be greatly reduced and the optimal values of decision variables are obtained faster.

       In order to demonstrate the advantages of GA, it was compared with particle swarm optimization (PSO) algorithm and simulated annealing algorithm (SAA). Next, the PSO algorithm and SAA were used for optimization. Through comparison, it was found that the GA can achieve lower warranty costs, and converges earlier with higher operational efficiency; therefore, it has more advantages in solving this model.

Round 2

Reviewer 3 Report

Dear Authors, thank you for reworking. Highly appreciated.

Reviewer 4 Report

The authors have addressed my previous concerns and significantly improved the paper quality. I do not have additional comments.